# $Q$-Pensieve: Boosting Sample Efficiency of Multi-Objective RL Through Memory Sharing of $Q$-Snapshots

**Wei Hung[1,2]\*, Bo-Kai Huang[1]\*, Ping-Chun Hsieh[1], Xi Liu[3]**

[1]Department of Computer Science, National Yang Ming Chiao Tung University, Hsinchu, Taiwan
[2]Research Center for Information Technology Innovation, Academia Sinica, Taipei, Taiwan
[3]Applied Machine Learning, Meta AI, Menlo Park, CA, USA
{hwei1048576.cs08,pinghsieh}@nycu.edu.tw, xiliu.tamu@gmail.com

## Abstract

Many real-world continuous control problems are in the dilemma of weighing the pros and cons, multi-objective reinforcement learning (MORL) serves as a generic framework of learning control policies for different preferences over objectives. However, the existing MORL methods either rely on multiple passes of explicit search for finding the Pareto front and therefore are not sample-efficient, or utilizes a shared policy network for coarse knowledge sharing among policies. To boost the sample efficiency of MORL, we propose $Q$-Pensieve, a policy improvement scheme that stores a collection of $Q$-snapshots to jointly determine the policy update direction and thereby enables data sharing at the policy level. We show that $Q$-Pensieve can be naturally integrated with soft policy iteration with convergence guarantee. To substantiate this concept, we propose the technique of $Q$ replay buffer, which stores the learned $Q$-networks from the past iterations, and arrive at a practical actor-critic implementation. Through extensive experiments and an ablation study, we demonstrate that with much fewer samples, the proposed algorithm can outperform the benchmark MORL methods on a variety of MORL benchmark tasks.

## 1 Introduction

Many real-world sequential decision-making problems involve the joint optimization of multiple objectives, while some of them may be in conflict. For example, in robot control, it is expected that the robot can run fast while consuming as little energy as possible; nevertheless, we inevitably need to use more energy to make the robot run fast, regardless of how energy-efficient the robot motion is. Moreover, various other real-world continuous control problems are also multi-objective tasks by nature, such as congestion control in communication networks (Ma et al., 2022) and diversified portfolios (Abdolmaleki et al., 2020). Moreover, the relative importance of these objectives could vary over time (Roijers and Whiteson, 2017). For example, the preference over energy and speed in robot locomotion could change with the energy budget; network service providers need to continuously switch service among various networking applications (e.g., on-demand video streaming versus real-time conferencing), each of which could have preferences over latency and throughput.

To address the above practical challenges, multi-objective reinforcement learning (MORL) serves as one classic and popular formulation for learning optimal control strategies from vector-valued reward signal and achieve favorable trade-off among the objectives. In the MORL framework, the goal is to learn a collection of policies, under which the attained return vectors recover as much of the Pareto front as possible. One popular approach to addressing MORL is to explicitly search for the Pareto front with an aim to maximize the hypervolume associated with the reward vectors, such as evolutionary search (Xu et al., 2020) and search by first-order stationarity (Kyriakis et al., 2022). While being effective, explicit search algorithms are known to be rather sample-inefficient as the data sharing among different passes of explicit search is rather limited. As a result, it is typically

---

\*Equal contribution.

difficult to maintain a sufficiently diverse set of optimal policies for different preferences within a reasonable number of training samples. Another way to address MORL is to implicitly search for non-dominated policies through linear scalarization, i.e., convert the vector-valued reward signal to a single scalar with the help of a linear preference and thereafter apply a conventional single-objective RL algorithm for iteratively improving the policies (e.g., (Abels et al., 2019; Yang et al., 2019)). To enable implicit search for diverse preferences simultaneously, a single network is typically used to express a whole collection of policies. As a result, some level of data sharing among policies of different preferences is done implicitly through the shared network parameters. However, such sharing is clearly not guaranteed to achieve policy improvement for all preferences. Therefore, there remains one critical open research question to be answered: *How to boost the sample efficiency of MORL through better policy-level knowledge sharing?*

To answer this question, we revisit MORL from the perspective of *memory sharing* among the policies learned across different training iterations and propose $\boldsymbol{Q}$-*Pensieve*, where a "*Pensieve*", as illustrated in the novel *Harry Potter*, is a magical device used to store pieces of personal memories, which can later be shared with someone else. By drawing an analogy between the memory sharing among humans and the knowledge sharing among policies, we propose to construct a $\boldsymbol{Q}$-Pensieve, which stores snapshots of the $\boldsymbol{Q}$-functions of the policies learned in the past iterations. Upon improving the policy for a specific preference, we expect that these $\boldsymbol{Q}$-snapshots could help jointly determine the policy update direction. In this way, we explicitly enforce knowledge sharing on the policy level and thereby enhance the sample use in learning optimal policies for various preferences. To substantiate this idea, we start by considering $Q$-Pensieve memory sharing in the tabular planning setting and integrate $Q$-Pensieve with the soft policy iteration for entropy-regularized MDPs. Inspired by (Yang et al., 2019), we leverage the envelope operation and propose the $\boldsymbol{Q}$-Pensieve policy iteration for MORL, which we show would preserve the similar convergence guarantee as the standard single-objective soft policy iteration. Based on this result, we propose a practical implementation that consists of two major components: (i) We introduce the technique of $\boldsymbol{Q}$ replay buffer. Similar to the standard replay buffer of state transitions, a $\boldsymbol{Q}$ replay buffer is meant to achieve sample reuse and improve sample efficiency, but notably at the policy level. Through the use of $\boldsymbol{Q}$ replay buffer, we can directly obtain a large collection of $\boldsymbol{Q}$ functions, each of which corresponds to a policy in a prior training iteration, without any additional efforts or computation in forming the $\boldsymbol{Q}$-Pensieve. (ii) We convert the $\boldsymbol{Q}$-Pensieve policy iteration into an actor-critic off-policy MORL algorithm by adapting the soft actor critic to the multi-objective setting and using it as the base of our implementation.

The main contributions of this paper can be summarized as:

- We identify the critical sample inefficiency issue in MORL and address this issue by proposing $\boldsymbol{Q}$-Pensieve, which is a policy improvement scheme for enhancing knowledge sharing on the policy level. We then present $\boldsymbol{Q}$-Pensieve policy iteration and establish its convergence property.

- We substantiate the concept of $\boldsymbol{Q}$-Pensieve policy iteration by proposing the technique of $\boldsymbol{Q}$ replay buffer and arrive at a practical actor-critic type practical implementation.

- We evaluate the proposed algorithm in various benchmark MORL environments, including Deep Sea Treasure and MuJoCo. Through extensive experiments and an ablation study, we demonstrate the the proposed $\boldsymbol{Q}$-Pensieve can indeed achieve significantly better empirical sample efficiency than the popular benchmark MORL algorithms, in terms of multiple common MORL performance metrics, including hypervolume and utility.

## 2 PRELIMINARIES

**Multi-Objective Markov Decision Processes (MOMDPs).** We consider the formulation of MOMDP defined by the tuple $(\mathcal{S}, \mathcal{A}, \mathcal{P}, \boldsymbol{r}, \gamma, \mathcal{D}, \mathfrak{S}_{\boldsymbol{\lambda}}, \Lambda)$, where $\mathcal{S}$ denotes the state space, $\mathcal{A}$ is the action space, $\mathcal{P} : \mathcal{S} \times \mathcal{A} \times \mathcal{S} \to [0, 1]$ is the transition kernel of the environment, $\boldsymbol{r} : \mathcal{S} \times \mathcal{A} \to [-r_{\max}, r_{\max}]^d$ is the vector-valued reward function with $d$ as the number of objectives, $\gamma \in (0, 1)$ is the discount factor, $\mathcal{D}$ is the initial state distribution, $\mathfrak{S}_{\boldsymbol{\lambda}} : \mathbb{R}^d \to \mathbb{R}$ is the scalarization function (under some preference vector $\boldsymbol{\lambda} \in \mathbb{R}^d$), and $\Lambda$ denotes the set of all preference vectors. In this paper, we focus on the *linear reward scalarization* setting, i.e., $\mathfrak{S}_{\boldsymbol{\lambda}}(\boldsymbol{r}) = \boldsymbol{\lambda}^\top \boldsymbol{r}(s, a)$, as commonly adopted in the MORL literature (Abels et al., 2019; Yang et al., 2019; Kyriakis et al., 2022). Without loss of generality, we let $\Lambda$ be the unit simplex. If $d = 1$, an MOMDP would degenerate to a standard MDP, and we simply use $r(s, a)$ to denote the scalar reward. At each time step $t \in \mathbb{N} \cup \{0\}$, the learner receives the

observation $s_t$, takes an action $a_t$, and receives a reward vector $\boldsymbol{r}_t$. We use $\pi : \mathcal{S} \to \Delta(\mathcal{A})$ to denote a stationary randomized policy, where $\Delta(\mathcal{A})$ denotes the set of all probability distributions over the action space. Let $\Pi$ be the set of all such policies.

**Single-Objective Entropy-Regularized RL.** In the standard framework of single-objective entropy-regularized RL (Haarnoja et al., 2017; 2018; Geist et al., 2019), the goal is to learn an optimal policy for an entropy-regularized MDP, where an entropy regularization term is augmented to the original reward function. For a policy $\pi \in \Pi$, the regularized value functions $V^\pi : \mathcal{S} \to \mathbb{R}$ and $Q^\pi : \mathcal{S} \times \mathcal{A} \to \mathbb{R}$ can be characterized through the regularized Bellman equations as

$$Q^\pi(s, a) = r(s, a) + \gamma \mathbb{E}_{s' \sim \mathcal{P}(\cdot|s,a)}[V^\pi(s')], \tag{1}$$

$$V^\pi(s) = \mathbb{E}_{a \sim \pi(\cdot|s)}[Q^\pi(s, a) - \alpha \log \pi(a|s)], \tag{2}$$

where $\alpha$ is a temperature parameter that specifies the relative importance of the entropy regularization term. In this setting, the goal is to learn an optimal policy $\pi^*$ such that $Q^{\pi^*}(s, a) \geq Q^\pi(s, a)$, for all $(s, a)$, for all $\pi \in \Pi$. An optimal policy can be obtained through soft policy iteration, which alternates between soft policy evaluation and soft policy improvement: (i) Soft policy evaluation: For a policy $\pi$, the soft $Q$-function of $\pi$ can be obtained by iteratively applying the corresponding soft Bellman backup operator $\mathcal{T}^\pi$ defined as

$$\mathcal{T}^\pi Q(s, a) = r(s, a) + \gamma \mathbb{E}_{s' \sim \mathcal{P}(\cdot|s,a)}[V(s')], \tag{3}$$

where $V(s') = \mathbb{E}_{a' \sim \pi(\cdot|s')}[Q(s', a') - \alpha \log(\pi(a' \mid s'))]$. (ii) Soft policy improvement: In each iteration $k$, the policy is updated towards an energy-based policy induced by the soft $Q$-function, i.e.,

$$\pi_{k+1} = \arg\min_{\pi' \in \tilde{\Pi}} \mathrm{D}_{\mathrm{KL}}\left(\pi'(\cdot \mid s) \,\middle\|\, \frac{\exp\left(\frac{1}{\alpha} Q^{\pi_k}(s, \cdot)\right)}{Z^{\pi_k}(s)}\right), \tag{4}$$

where $\tilde{\Pi}$ is the set of parameterized policies of interest and $Z^{\pi_k}$ is the normalization term.

**Multi-Objective Entropy-Regularized RL.** We extend the standard single-objective RL with entropy regularization to the multi-objective setting. For each policy $\pi \in \Pi$, we define the multi-objective regularized value functions via the following multi-objective version of entropy-regularized Bellman equations as follows:

$$\boldsymbol{Q}^\pi(s, a) = \boldsymbol{r}(s, a) + \gamma \mathbb{E}_{s' \sim \mathcal{P}(\cdot|s,a)}[\boldsymbol{V}^\pi(s')], \tag{5}$$

$$\boldsymbol{V}^\pi(s) = \mathbb{E}_{a \sim \pi(\cdot|s)}[\boldsymbol{Q}^\pi(s, a) - \alpha \log \pi(a|s)\mathbf{1}_d], \tag{6}$$

where $\mathbf{1}_d$ denotes a $d$-dimensional vector of all ones.

In this paper, our goal is to learn a preference-dependent policy $\pi(\cdot|\cdot; \boldsymbol{\lambda})$ such that for any preference $\boldsymbol{\lambda} \in \Lambda$, $\boldsymbol{\lambda}^\top \boldsymbol{Q}^{\pi(\cdot|\cdot;\boldsymbol{\lambda})}(s, a; \boldsymbol{\lambda}) \geq \boldsymbol{\lambda}^\top \boldsymbol{Q}^{\pi'}(s, a)$, for all $(s, a)$, for all $\pi' \in \Pi$. For ease of notation, we let $\boldsymbol{V}^{\pi(\cdot|\cdot;\boldsymbol{\lambda})}(s; \boldsymbol{\lambda}) \equiv \boldsymbol{V}^\pi(s; \boldsymbol{\lambda})$ and $\boldsymbol{Q}^{\pi(\cdot|\cdot;\boldsymbol{\lambda})}(s, a; \boldsymbol{\lambda}) \equiv \boldsymbol{Q}^\pi(s, a; \boldsymbol{\lambda})$ in the sequel.

## 3 ALGORITHMS

In this section, we propose our $\boldsymbol{Q}$-Pensieve learning algorithm for boosting the sample efficiency of multi-objective RL. We first describe the idea of $\boldsymbol{Q}$-Pensieve in the tabular planning setting by introducing $\boldsymbol{Q}$-Pensieve soft policy iteration. We then extend the idea to develop a practical deep reinforcement learning algorithm.

### 3.1 NAIVE MULTI-OBJECTIVE SOFT POLICY ITERATION

To solve MORL in the entropy-regularized setting, one straightforward approach is to leverage the single-objective soft policy improvement with the help of linear scalarization. That is, in each iteration $k$, the policy can be updated by

$$\pi_{k+1}(\cdot, \cdot; \boldsymbol{\lambda}) = \arg\min_{\pi' \in \tilde{\Pi}} \mathrm{D}_{\mathrm{KL}}\left(\pi'(\cdot \mid s) \,\middle\|\, \frac{\exp\left(\frac{1}{\alpha} \boldsymbol{\lambda}^\top \boldsymbol{Q}^{\pi_k}(s, \cdot; \boldsymbol{\lambda})\right)}{Z_{\boldsymbol{\lambda}}^{\pi_k}(s)}\right). \tag{7}$$

While (7) serves as a reasonable approach, designing a learning algorithm based on the update scheme in (7) could suffer from *sample inefficiency* due to the *lack of policy-level knowledge sharing*: In (7), the policy for each preference $\boldsymbol{\lambda}$ is updated completely separately based solely on the $\boldsymbol{Q}$-function under $\boldsymbol{\lambda}$. Moreover, as the update (7) relies on an accurate estimate of the $\boldsymbol{Q}$-function, the critic learning for the policy of each individual preference would typically require at least a moderate number of samples. These issues could be particularly critical for a large preference set in practice. While the use of a conditioned policy network (e.g., (Abels et al., 2019)), a commonly-used network architecture in the MORL literature, could somewhat mitigate this issue, it remains unclear whether the knowledge sharing induced by the conditioned network can indeed achieve policy improvement across various preferences. As a result, a systematic approach is needed for boosting the sample efficiency in MORL.

## 3.2 $Q$-PENSIEVE SOFT POLICY ITERATION

To boost the sample efficiency of MORL, we propose to enhance the policy-level knowledge sharing by constructing a $\boldsymbol{Q}$-Pensieve for *memory sharing across iterations*. Specifically, a $\boldsymbol{Q}$-Pensieve is a collection of $\boldsymbol{Q}$-snapshots obtained from the past iterations, and it is formed to boost the policy improvement update with respect to the $\boldsymbol{Q}$-function in the current iteration as these $\boldsymbol{Q}$-snapshots could offer potentially better policy improvement directions under linear scalarization. Moreover, one major computational benefit of $\boldsymbol{Q}$-Pensieve is that these $\boldsymbol{Q}$-snapshots are obtained without the need for any updates or additional samples from the environment (and hence are for free) as they already exist during training. We substantiate this idea by first introducing the $\boldsymbol{Q}$-Pensieve soft policy iteration in the tabular setting (i.e., $|\mathcal{S}|$ and $|\mathcal{A}|$ are finite) as follows:

**$Q$-Pensieve Policy Improvement.** In the policy improvement step of the $k$-th iteration, for each specific $\boldsymbol{\lambda}$, we update the policy as

$$\pi_{k+1}(\cdot|\cdot;\boldsymbol{\lambda}) = \arg\min_{\pi'\in\tilde{\Pi}} \mathrm{D}_{\mathrm{KL}}\left(\pi'\left(\cdot \mid s;\boldsymbol{\lambda}\right) \left\| \frac{\exp\left(\sup_{\boldsymbol{\lambda}'\in W_k(\boldsymbol{\lambda}),\boldsymbol{Q}'\in\mathcal{Q}_k} \frac{1}{\alpha}\boldsymbol{\lambda}^\top \boldsymbol{Q}'(s,\cdot;\boldsymbol{\lambda}')\right)}{Z_{\mathcal{Q}_k}(s)}\right.\right), \quad (8)$$

where $Z_{\mathcal{Q}_k}$ is again the normalization term, $W_k(\boldsymbol{\lambda}) \subset \Lambda$ is a set of preference vectors, and $\mathcal{Q}_k$ is a set of $\boldsymbol{Q}$-snapshots. The two sets $W_k(\boldsymbol{\lambda})$ and $\mathcal{Q}_k$ are to be selected as follows:

- For $W_k(\boldsymbol{\lambda})$, the only requirement is that $\boldsymbol{\lambda} \in W_k(\boldsymbol{\lambda})$, for all $k$. The preference sets can be different in different iterations.

- Similarly, for $\mathcal{Q}_k$, the only requirement is that $\boldsymbol{Q}^{\pi_k} \in \mathcal{Q}_k$, for all $k$. The set of $\boldsymbol{Q}$-snapshots can also be different in different iterations. Hence, the choice of $\mathcal{Q}_k$ is rather flexible.

When choosing $W_k(\boldsymbol{\lambda}) = \{\boldsymbol{\lambda}\}$ and $\mathcal{Q}_k = \{\boldsymbol{Q}^{\pi_k}\}$, one would recover the update in (7).

**Policy Evaluation.** In the policy evaluation step, we evaluate the policy that corresponds to each preference $\boldsymbol{\lambda}$ by iteratively applying the multi-objective softmax Bellman backup operator $\mathcal{T}_{\mathrm{MO}}^\pi$ as

$$(\mathcal{T}_{\mathrm{MO}}^\pi \boldsymbol{Q})(s,a;\boldsymbol{\lambda}) = \boldsymbol{r}(s,a) + \gamma\mathbb{E}_{s'\sim\mathcal{P}(\cdot|s,a),a'\sim\pi(\cdot|s';\boldsymbol{\lambda})}[\boldsymbol{Q}(s',a';\boldsymbol{\lambda}) - \alpha\log\pi(a'|s';\boldsymbol{\lambda})\mathbf{1}_d]. \quad (9)$$

**Remark 1** The $\boldsymbol{Q}$-Pensieve update in (8) is inspired by the envelope Q-learning (EQL) technique (Yang et al., 2019), where in each iteration $k$, the Q-learning update takes into account the envelope formed by the $\boldsymbol{Q}$-functions of the current policy $\pi_k$ for different preferences. The fundamental difference between $\boldsymbol{Q}$-Pensieve and EQL is that $\boldsymbol{Q}$-Pensieve further achieves *memory sharing across training iterations* through the use of $\boldsymbol{Q}$-snapshots from the past iterations, and EQL focuses mainly on the use of the $\boldsymbol{Q}$-function of the current iteration.

**Convergence of $Q$-Pensieve Soft Policy Iteration.** Another nice feature of the $\boldsymbol{Q}$-Pensieve policy improvement step is that it preserves the similar convergence result as the standard single-objective soft policy iteration, as stated below. The proof of Theorem 3.1 is provided in Appendix A.

**Theorem 3.1** *Under the $\boldsymbol{Q}$-Pensieve soft policy iteration given by (8) and (9), the sequence of preference-dependent policies $\{\pi_k\}$ converges to a policy $\pi^*$ such that $\boldsymbol{\lambda}^\top\boldsymbol{Q}^{\pi^*}(s,a;\boldsymbol{\lambda}) \geq \boldsymbol{\lambda}^\top\boldsymbol{Q}^\pi(s,a)$ for all $\pi \in \Pi$, for all $(s,a) \in \mathcal{S}\times\mathcal{A}$ and for all $\boldsymbol{\lambda} \in \Lambda$.*

### 3.3 PRACTICAL IMPLEMENTATION OF $Q$-PENSIEVE

In this section, we present the implementation of proposed $\boldsymbol{Q}$-Pensieve algorithm for learning policies with function approximation for the general state and action spaces.

$\boldsymbol{Q}$ **Replay Buffer.** Based on (8), we know that the policy update of $\boldsymbol{Q}$-Pensieve would involve both the current $\boldsymbol{Q}$-function and the $\boldsymbol{Q}$-snapshots from the past iterations. To implement this, we introduce $\boldsymbol{Q}$ replay buffer, which could store multiple $\boldsymbol{Q}$-networks in a predetermined manner (e.g., first-in first-out). Notably, unlike the conventional experience replay buffer (Mnih et al., 2013) of state transitions, $\boldsymbol{Q}$ replay buffer stores the learned $\boldsymbol{Q}$-networks in past iterations as candidates for forming the $\boldsymbol{Q}$-Pensieve. On the other hand, while each $\boldsymbol{Q}$-network would require a moderate amount of memory usage, we found that in practice a rather small $\boldsymbol{Q}$ replay buffer is already effective enough for boosting the sample efficiency. We further illustrate this observation through the experimental results in Section 4.

Next, we convert the $\boldsymbol{Q}$-Pensieve soft policy iteration into an actor-critic off-policy MORL algorithm. Specifically, we adapt the idea of soft actor critic to $\boldsymbol{Q}$-Pensieve by minimizing the residual of the multi-objective soft $\boldsymbol{Q}$-function: Let $\theta$ and $\phi$ be the parameters of the policy network and the critic network, respectively. Then, the critic network is updated by minimizing the following loss

$$\mathcal{L}_{\boldsymbol{Q}}(\phi;\boldsymbol{\lambda}) = \mathbb{E}_{(s,a)\sim\mu}\left[\boldsymbol{\lambda}^{\top}\left(\boldsymbol{Q}_{\phi}(s,a;\boldsymbol{\lambda}) - \left(\boldsymbol{r}(s,a) + \gamma\mathbb{E}_{s'\sim\mathcal{P}(\cdot|s,a)}\left[\boldsymbol{V}_{\bar{\phi}}(s')\right]\right)\right)^2\right], \quad (10)$$

where $\bar{\phi}$ is the parameter of the target network and $\mu$ is the sampling distribution of the state-action pairs (e.g., a distribution induced by a replay buffer of state transitions). On the other hand, based on (8), the policy network is updated by minimizing the following objective

$$\mathcal{L}_{\pi}(\theta;\boldsymbol{\lambda}) = \mathbb{E}_{s\sim\mu}\left[\mathbb{E}_{a\sim\pi_{\theta}}\left[\sup_{\boldsymbol{\lambda}'\in W(\boldsymbol{\lambda}),\boldsymbol{Q}'\in\mathcal{Q}}\left\{\alpha\log\left(\pi_{\theta}(a\mid s;\boldsymbol{\lambda})\right) - \boldsymbol{\lambda}^{\intercal}\boldsymbol{Q}'(s,a;\boldsymbol{\lambda}')\right\}\right]\right]. \quad (11)$$

The overall architecture of $Q$-Pensieve is provided in Figure 1. The pseudo code of the $Q$-Pensieve algorithm is described in Algorithm 1 in Appendix. The code of our experiments is available [1]. Notably, in Section 4 we show that empirically a relatively small Q buffer size (e.g., 4 in our experiments) can already offer a significant performance improvement.

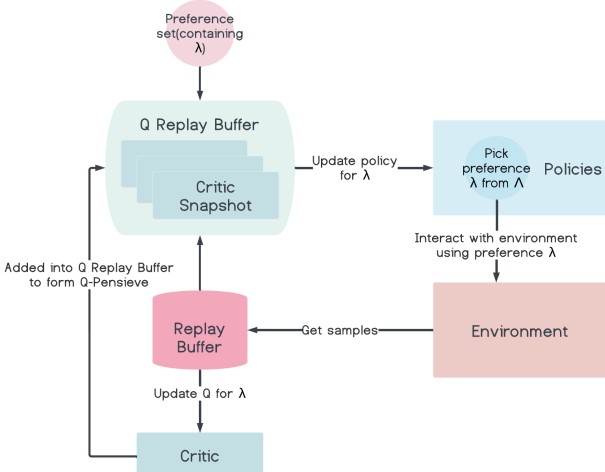

Figure 1: The architecture of $\boldsymbol{Q}$-Pensieve.

## 4 EXPERIMENTS

In this section, we demonstrate the effectiveness of $\boldsymbol{Q}$-Pensieve on various benchmark RL tasks and discuss how $\boldsymbol{Q}$-Pensieve boosts the sample efficiency through an extensive ablation study.

---

[1]https://github.com/NYCU-RL-Bandits-Lab/Q-Pensieve

## 4.1 EXPERIMENTAL CONFIGURATION

**Popular Benchmark Methods.** We compare the proposed algorithm against various popular benchmark methods, including the Conditioned Network with Diverse Experience Replay (CN-DER) in (Abels et al., 2019), the Prediction-Guided Multi-Objective RL (PGMORL) in (Xu et al., 2020), the Pareto Following Algorithm (PFA) in (Parisi et al., 2014), and SAC (Haarnoja et al., 2018). For CN-DER, as the original CN-DER is built on deep Q-networks (DQN) for discrete actions, we modify the source code of Abels et al. (2019) for continuous control by implementing CN-DER on top of DDPG. Moreover, we follow the same DER technique, which uses a diverse replay buffer and gives priority according to how much the samples increase the overall diversity of the buffer. For PGMORL and PFA, we use the open-source implementation of (Xu et al., 2020) for the experiments. As these explicit search methods typically require more samples before reaching a comparable performance level, we evaluate the performance PGMORL and PFA under both 1 times and $\beta$ times ($\beta > 1$) of the number of samples used by $\boldsymbol{Q}$-Pensieve to demonstrate the sample efficiency of $\boldsymbol{Q}$-Pensieve. For SAC, as the MORL problem reduces a single-objective one under a fixed preference, we train multiple models using single-objective SAC (one model for each fixed preference) as a performance reference for other MORL methods.

**Performance Metrics.** In the evaluation, we consider the following three commonly-used performance metrics for MORL:

- **HyperVolume (HV)**: Let $\mathcal{R}$ be a set of return vectors attained and $\boldsymbol{r}_0 \in \mathbb{R}^d$ be a reference point. Then, we define the HyperVolume as $\mathrm{HV} := \int_{H(\mathcal{R})} \mathbb{I}\{z \in H(\mathcal{R})\} dz$, where $H(\mathcal{R}) := \left\{z \in \mathbb{R}^d : \exists \boldsymbol{r} \in \mathcal{R}, \boldsymbol{r}_0 \prec z \prec \boldsymbol{r}\right\}$ and $\mathbb{I}$ is the indicator function.
- **Utility (UT)**: To further evaluate the performance under linear scalarization, we define the utility metric as $\mathrm{UT} := \mathbb{E}_{\boldsymbol{\lambda}}\left[\sum_{t=0}^T \boldsymbol{\lambda}^\top \boldsymbol{r}_t\right]$, where the preference $\boldsymbol{\lambda}$ is sampled uniformly from $\Lambda$.
- **Episodic Dominance (ED)**: To compare the performance of a pair of algorithms, we define Episodic Dominance as $\mathrm{ED}_{1,2} := \mathbb{E}_{\boldsymbol{\lambda}}[\mathbb{I}\{\sum_{t=0}^{T_1} \boldsymbol{\lambda}^\top \boldsymbol{r}_t^1 > \sum_{t=0}^{T_2} \boldsymbol{\lambda}^\top \boldsymbol{r}_t^2\}]$, where $\boldsymbol{r}_t^1$, $\boldsymbol{r}_t^2$ are the return vectors, and $T_1, T_2$ are the episode lengths of algorithm 1 and 2, respectively. ED serves as a useful metric for pairwise comparison in those problems where the return vectors under different preferences can differ by a lot in magnitude (in this case, HV and UT could be dominated by the return vectors of a few preferences).

**Evaluation Domains.** We evaluate the algorithms in the following domains: (i) Continuous Deep Sea Treasure (DST): a two-objective continuous control task modified from the original DST environment. (ii) Multi-Objective Continuous LunarLander: a four-objective task modified from the classic control task in the OpenAI gym. (iii) Multi-Objective MuJoCo: modified benchmark locomotion tasks with either two or three objectives.

**Configuration of $\boldsymbol{Q}$-Pensieve.** For $\boldsymbol{Q}$-Pensieve, at each policy update, we set the size of the preference set $W_k(\boldsymbol{\lambda})$ to be 5 (including $\boldsymbol{\lambda}$ and another four preferences drawn randomly) and set the size of the $\boldsymbol{Q}$ replay buffer to be 4, unless stated otherwise.

## 4.2 EXPERIMENTAL RESULTS

**Does $\boldsymbol{Q}$-Pensieve achieve better sample efficiency than the MORL benchmark methods?** Table 1 shows the performance of $\boldsymbol{Q}$-Pensieve and the benchmark methods in terms of the three metrics. For each algorithm, we report the mean and the standard deviation over five random seeds. We can observe that $\boldsymbol{Q}$-Pensieve consistently enjoys higher HV, UT, and ED in almost all the domains. More importantly, $\boldsymbol{Q}$-Pensieve indeed exhibits superior sample efficiency as it still outperforms the explicit search methods (i.e., PFA and PGMORL) even if these methods are given 10 times of the number of samples used by $\boldsymbol{Q}$-Pensieve. Moreover, we can observe that the explicit search methods (i.e., PFA and PGMORL) often have larger HV than the implicit search method (such as CN-DER), while implicit search methods tend to have larger UT. This manifests the design principles and the characteristics of the two families of approaches, where explicit search is designed mainly for achieving large HV and implicit search typically aims for larger scalarized return.

**How much improvement in sample efficiency can $\boldsymbol{Q}$-Pensieve achieve compared to training multiple single-objective SAC models separately?** To answer this question, we conduct experiments on 2-objective MuJoCo tasks and consider a whole range of 19 preference vectors

Table 1: Comparison of $Q$-Pensieve and other benchmark algorithms in terms of the three metrics across ten domains. We report the mean and standard deviation over five random seeds. The ED is calculated through comparing each algorithm to a multi-objective version of SAC (equivalent to $Q$-Pensieve with the size of the preference set equal to 1 and without $Q$ replay buffer). We set $\beta = 10$ for HalfCheetah2d, Ant2d, Ant3d, and Hopper3d, set $\beta = 5$ for LunarLander4d, LunarLander5d, and Hopper5d, and set $\beta = 3$ for DST2d, Hopper2d, and Walker2d.

| Environments | Metrics | PFA (1.5M steps) | PFA ($1.5 \times \beta$M steps) | PGMORL (1.5M steps) | PGMORL ($1.5 \times \beta$M steps) | CN-DER (1.5M steps) | $Q$-Pensieve (1.5M steps) |
|---|---|---|---|---|---|---|---|
| DST2d | HV($\times 10^2$) | 7.43±3.68 | 8.67±1.49 | 8.10±1.57 | 8.13±1.61 | 5.36±4.71 | **10.21±1.40** |
| | UT($\times 10^0$) | -9.27±6.03 | -6.86±6.06 | 4.90±0.44 | 5.02±0.35 | -5.10±15.73 | **7.31±0.91** |
| | ED | 0.13±0.11 | 0.10±0.08 | 0.25±0.18 | 0.28±0.18 | 0.21±0.17 | **0.54±0.11** |
| LunarLander4d | HV($\times 10^8$) | - | - | 0.32±0.11 | 0.38±0.11 | 1.50±0.60 | **2.10±0.10** |
| | UT($\times 10^1$) | - | - | -0.26±0.27 | 1.10±0.50 | 3.60±2.90 | **5.10±0.30** |
| | ED | - | - | 0.02±0.01 | 0.04±0.04 | 0.21±0.12 | **0.49±0.05** |
| LunarLander5d | HV($\times 10^{11}$) | - | - | 1.81±0.20 | 1.87±0.42 | 8.64±0.15 | **9.48±1.84** |
| | UT($\times 10^1$) | - | - | -2.77±0.68 | -4.38±1.02 | 0.56±0.42 | **1.07±0.24** |
| | ED | - | - | 0.05±0.02 | 0.05±0.02 | 0.49±0.01 | **0.52±0.02** |
| HalfCheetah2d | HV($\times 10^7$) | 0.73±0.19 | 1.31±0.26 | 0.53±0.17 | 0.28±0.29 | 2.08±0.54 | **3.82±0.27** |
| | UT($\times 10^3$) | 0.31±0.20 | 1.02±0.40 | -0.28±0.94 | 0.09±0.17 | 5.09±3.57 | **5.61±0.31** |
| | ED | 0.08±0.10 | 0.10±0.06 | 0.01±0.00 | 0.11±0.05 | 0.02±0.01 | **0.54±0.08** |
| Hopper2d | HV($\times 10^6$) | 0.49±0.46 | 1.01±0.62 | 0.63±0.48 | 1.31±0.48 | 0.56±0.16 | **1.33±0.20** |
| | UT($\times 10^2$) | 2.89±1.93 | 3.50±1.85 | 1.94±2.46 | 3.70±1.78 | 1.42±1.00 | **4.08±1.10** |
| | ED | 0.31±0.17 | 0.41±0.10 | 0.31±0.25 | 0.31±0.11 | 0.04±0.03 | **0.43±0.09** |
| Hopper3d | HV($\times 10^9$) | - | - | 0.29±0.37 | 0.91±1.39 | 3.70±0.81 | **9.56±0.60** |
| | UT($\times 10^3$) | - | - | 0.19±0.16 | 0.31±0.26 | 0.72±0.16 | **1.39±0.15** |
| | ED | - | - | 0.02±0.03 | 0.03±0.03 | 0.07±0.03 | **0.55±0.08** |
| Hopper5d | HV($\times 10^{13}$) | - | - | 0.63±0.11 | 0.43±0.09 | 3.42±0.93 | **7.24±0.31** |
| | UT($\times 10^2$) | - | - | 1.48±0.28 | 1.63±0.21 | 1.76±0.43 | **3.37±0.65** |
| | ED | - | - | 0.18±0.07 | 0.14±0.05 | 0.21±0.05 | **0.52±0.05** |
| Ant2d | HV($\times 10^6$) | 0.17±0.05 | 0.77±0.53 | 0.14±0.03 | 0.13±0.04 | 5.03±3.60 | **10.01±1.86** |
| | UT($\times 10^2$) | -0.06±0.01 | 0.14±0.14 | -0.21±0.15 | -0.18±0.38 | 3.68±2.34 | **14.04±3.03** |
| | ED | 0.22±0.03 | 0.22±0.02 | 0.21±0.02 | 0.21±0.03 | 0.21±0.08 | **0.60±0.07** |
| Ant3d | HV($\times 10^8$) | - | - | 0.41±0.48 | 0.68±0.62 | 13.00±4.11 | **21.87±1.07** |
| | UT($\times 10^3$) | - | - | 0.18±0.05 | 0.25±0.05 | 0.49±0.23 | **1.14±0.22** |
| | ED | - | - | 0.02±0.02 | 0.03±0.03 | 0.28±0.14 | **0.56±0.07** |
| Walker2d | HV($\times 10^6$) | 0.52±0.20 | 1.05±0.44 | 0.83±0.42 | **1.28±0.66** | 0.42±0.09 | 1.12±0.36 |
| | UT($\times 10^2$) | 0.23±0.13 | 0.95±0.55 | 0.38±0.24 | 1.20±0.67 | 3.17±0.53 | **6.37±1.42** |
| | ED | 0.32±0.06 | 0.37±0.09 | 0.30±0.10 | 0.34±0.12 | 0.21±0.11 | **0.48±0.10** |

$([0.05, 0.95], [0.1, 0.9], [0.15, 0.85], \cdots, [0.95, 0.05])$. We train 19 models by using single-objective SAC, one model for each individual preference. Each model is trained for 1.5M steps (and hence the total number of steps under SAC is 28.5M steps). By contrast, $Q$-Pensieve only uses 1.5M steps in total in learning policies for all the preferences. Figure 2 shows the return vectors attained by $Q$-Pensieve and the collection of 19 SAC models. $Q$-Pensieve can achieve comparable or better returns than the collection of SAC models with only $1/19$ of the samples. This further demonstrate the sample efficiency of $Q$-Pensieve.

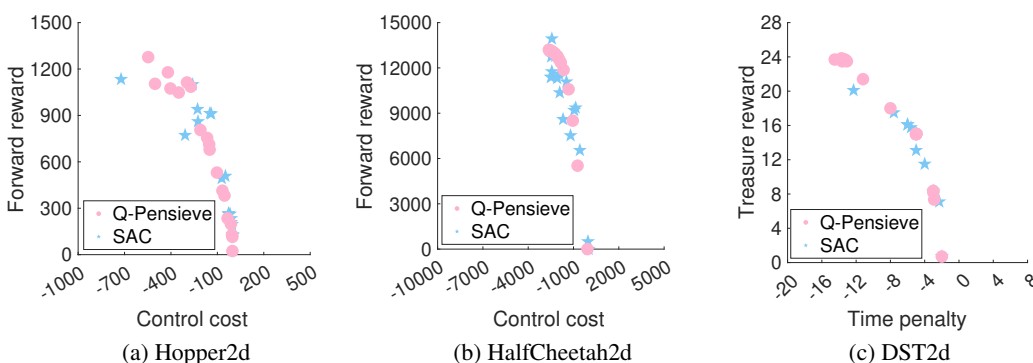

Figure 2: Return vectors attained by $Q$-Pensieve and the collection of single-objective SAC models under 19 preferences.

**Why can $Q$-Pensieve outperform single-objective SAC in some cases?** From Figures 2(a) and (c), we see that $Q$-Pensieve can attain some return vectors that are strictly better than those of the single-objective SAC models. The reasons behind this phenomenon are minaly two-fold: (i) Under single-objective SAC, despite that we train one model for each individual preference, it could still occur that single-objective SAC gets stuck at a sub-optimal policy under some preferences. (ii) By contrast, $Q$-Pensieve has a better chance of escaping from these sub-optimal policies with the help of the $Q$-snapshots in the $Q$ replay buffer.

To verify the above argument, we design a hybrid SAC algorithm as follows: (a) For the first $10^5$ time steps, this algorithm simply follows the single-objective SAC. (b) At time step $10^5$, it switches to the update rule of $Q$-Pensieve based on the $Q$-snapshots stored in the $Q$ replay buffer of another model trained under $Q$-Pensieve algorithm in parallel. Figure 3 shows the performance of this hybrid algorithm in DST and HalfCheetah. Clearly, the $Q$-Pensieve update could help the SAC model escape from the sub-optimal policies, under various preferences.

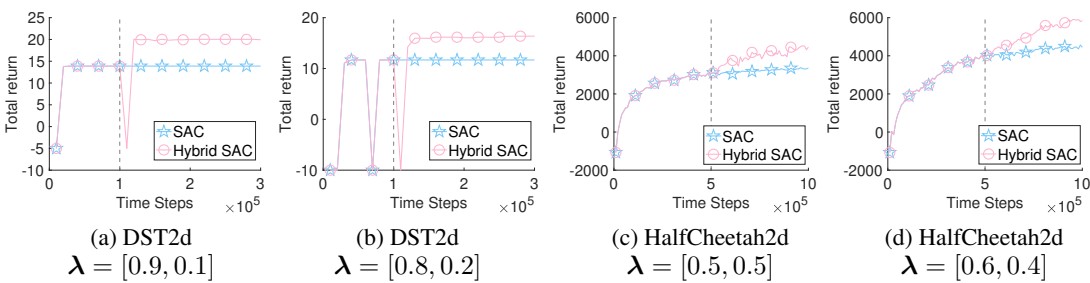

(a) DST2d
$\boldsymbol{\lambda} = [0.9, 0.1]$

(b) DST2d
$\boldsymbol{\lambda} = [0.8, 0.2]$

(c) HalfCheetah2d
$\boldsymbol{\lambda} = [0.5, 0.5]$

(d) HalfCheetah2d
$\boldsymbol{\lambda} = [0.6, 0.4]$

Figure 3: Comparison of standard single-objective SAC and the hybrid SAC assisted by another $Q$-Pensieve model trained in parallel.

**An ablation study on $Q$ replay buffer.** To verify the effectiveness of the technique of $Q$ replay buffer, we compare the performance of $Q$-Pensieve with buffer size equal to $4$ and that without using $Q$ replay buffer (termed "Vanilla" in Figures 4 and 5). Figure 4 and 5 show the attained return vectors and HV of both methods. We can see that $Q$ replay buffer indeed leads to a better policy improvement behavior, in terms of both HV and the scalarized returns. However, these figures may sometimes oscillate a lot in the end period. It is because our algorithm finds solutions from another $Q$-vector, and their inner product of $Q$ and preference may be quite close. We can check the points are in the same contour.

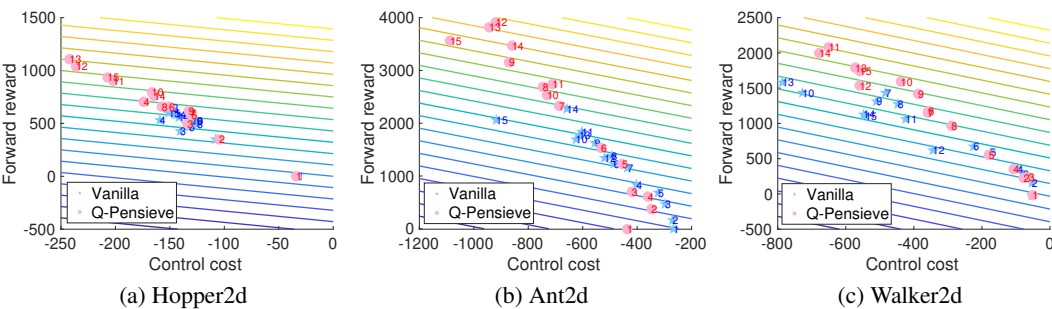

(a) Hopper2d

(b) Ant2d

(c) Walker2d

Figure 4: Return vectors attained under preference $\boldsymbol{\lambda} = [0.5, 0.5]$ at different training stages (We also plot return vectors under others preference in Figure 7 and Figure 8 in Appendix). A number $x$ on the red or blue marker indicates that the model is obtained at $100 \cdot x$ thousand steps.

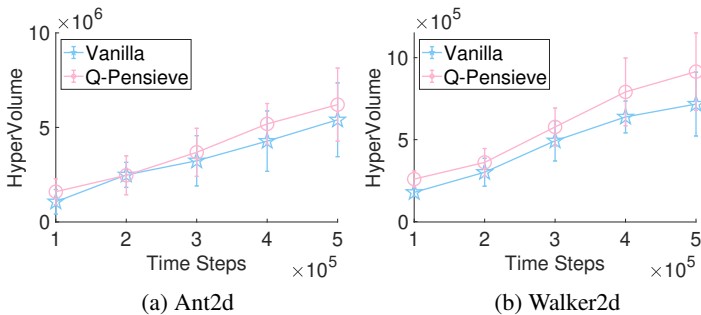

(a) Ant2d    (b) Walker2d

Figure 5: A comparison in HV between $Q$-Pensieve with buffer size equal to 4 and that without using $Q$ replay buffer at different training stages.

## 5 RELATED WORK

The multi-objective RL problems have been extensively studied from two major perspectives:

**Explicit Search.** A plethora of prior works on MORL updates a policy or a set of policies by explicitly searching for the Pareto front of the reward space. To learn policies under time-varying preferences, (Natarajan and Tadepalli, 2005) presented to store a set of policies, which are to be used in searching for a proper policy for a new preference without learning from scratch. (Lizotte et al., 2012) leveraged linear value function approximation to search for optimal policies. (Van Moffaert and Nowé, 2014) proposed Pareto Q-learning, which stores the immediate rewards and the non-dominated future return vectors separately and leverage the Pareto dominance for selecting the actions in $Q$-learning. (Parisi et al., 2014) presented a policy gradient approach to search for non-dominated policies. (Mossalam et al., 2016) solves MORL via scalarized $Q$-learning along with the concept of prioritizing the corner weights for selecting the preference of the scalarized problem. (Xu et al., 2020) proposed an evolutionary approach to search for the Pareto set of policies, with the help of a prediction model for determining the search direction. (Kyriakis et al., 2022) presented a policy gradient method by approximating the Pareto front via a first-order necessary condition. However, the above explicit search algorithms are known to be rather sample-inefficient as the knowledge sharing among different passes of search is limited.

**Implicit Search.** Another class of algorithms are designed to improve policies for multiple preferences through implicit search. For example, (Abels et al., 2019) presents Conditioned Network, which extends the standard single-objective DQN to learning preference-dependent multi-objective $Q$-functions. To achieve scale-invariant MORL, (Abdolmaleki et al., 2020) proposed to first learn the $Q$-functions for different objectives and encode the preference through constraints. Recently, (Yang et al., 2019) proposes envelope $Q$-functions to encourage knowledge sharing among the Q functions of different the current multi-objective $Q$-values that any policy can benefit from other preferences' experiences, that make training more efficiently, and (Zhou et al., 2020) proposed model-based envelope value iteration base on envelope $Q$-functions, it provides an efficient way to get optimal multi-objective $Q$-functions. Despite that our method is inspired by (Yang et al., 2019), the main difference between our work and theirs is that we boost the sample efficiency of MORL via explicit memory sharing among policies learned during training.

## 6 CONCLUSION

This paper proposes $Q$-Pensieve, which significantly enhances the policy-level data sharing through in order to boost the sample efficiency of MORL problems. We substantiate the idea by presenting $Q$-Pensieve soft policy iteration in the tabular setting and show that it preserves the global convergence property. Then, to implement the $Q$-Pensieve policy improvement step, we introduce the $Q$ replay buffer technique, which offers a simple yet effective way to maintain $Q$-snapshot. Our experiments demonstrate that $Q$-Pensieve is a promising approach in that it can outperform the state-of-the-art MORL methods with much fewer samples in a variety of MORL benchmark tasks.

## ACKNOWLEDGMENTS

This material is based upon work partially supported by the National Science and Technology Council (NSTC), Taiwan under Contract No. 110-2628-E-A49-014 and Contract No. 111-2628-E-A49-019, and based upon work partially supported by the Higher Education Sprout Project of the National Yang Ming Chiao Tung University and Ministry of Education (MOE), Taiwan.

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

APPENDIX

## A    PROOF OF THEOREM 3.1

Before proving Theorem 3.1, we first present two supporting lemmas as follows. To begin with, we establish the policy improvement property of the $\boldsymbol{Q}$-Pensieve update. Recall that the $\boldsymbol{Q}$-Pensieve policy update is that for each preference $\boldsymbol{\lambda} \in \Lambda$,

$$
\pi_{k+1}(\cdot|s;\boldsymbol{\lambda}) = \arg\min_{\pi'\in\Pi} \underbrace{D_{\text{KL}}\left(\pi'\left(\cdot\mid s;\boldsymbol{\lambda}\right) \,\middle\|\, \frac{\exp\left(\sup_{\boldsymbol{\lambda}'\in W_k(\boldsymbol{\lambda}),\boldsymbol{Q}'\in\mathcal{Q}_k} \frac{1}{\alpha}\boldsymbol{\lambda}^\top \boldsymbol{Q}'(s,\cdot;\boldsymbol{\lambda}')\right)}{Z_{\mathcal{Q}}(s)}\right)}_{=:L(\pi';\boldsymbol{\lambda})}.
$$

$$(12)$$

**Lemma 1 ($\boldsymbol{Q}$-Pensieve Policy Improvement)** *Under the $\boldsymbol{Q}$-Pensieve policy improvement update, we have $\boldsymbol{\lambda}^\top \boldsymbol{Q}^{\pi_k}(s,a;\boldsymbol{\lambda}) \leq \boldsymbol{\lambda}^\top \boldsymbol{Q}^{\pi_{k+1}}(s,a;\boldsymbol{\lambda})$, for all state-action pairs $(s,a) \in \mathcal{S}\times\mathcal{A}$, for all preference vectors $\boldsymbol{\lambda} \in \Lambda$, and for all iteration $k \in \mathbb{N}\cup\{0\}$.*

**Proof (Lemma 1)** By the update rule in (12), we know that $\pi_{k+1}$ is a minimizer of $L(\pi';\boldsymbol{\lambda})$ and hence $L(\pi_{k+1};\boldsymbol{\lambda}) \leq L(\pi_k;\boldsymbol{\lambda})$. This implies that for each state $s\in\mathcal{S}$, we have

$$
\mathbb{E}_{a\sim\pi_{k+1}(\cdot|s)}\left[\boldsymbol{\lambda}^\top\mathbf{1}_d\cdot\log\pi_{k+1}(a|s;\boldsymbol{\lambda}) - \frac{1}{\alpha}\sup_{\boldsymbol{\lambda}'\in W_k(\boldsymbol{\lambda}),\boldsymbol{Q}'\in\mathcal{Q}_k}\boldsymbol{\lambda}^\top\boldsymbol{Q}'(s,a;\boldsymbol{\lambda}') + \log Z_{\mathcal{Q}_k}(s)\right]
$$

$$
\leq \mathbb{E}_{a\sim\pi_k(\cdot|s)}\left[\boldsymbol{\lambda}^\top\mathbf{1}_d\cdot\log\pi_k(a|s;\boldsymbol{\lambda}) - \frac{1}{\alpha}\sup_{\boldsymbol{\lambda}'\in W_k(\boldsymbol{\lambda}),\boldsymbol{Q}'\in\mathcal{Q}_k}\boldsymbol{\lambda}^\top\boldsymbol{Q}'(s,a;\boldsymbol{\lambda}') + \log Z_{\mathcal{Q}_k}(s)\right]. \quad (13)
$$

Since $Z_{\mathcal{Q}_k}$ only depends on the state, the inequality (13) reduces to

$$
\mathbb{E}_{a\sim\pi_{k+1}(\cdot|s)}\left[\boldsymbol{\lambda}^\top\mathbf{1}_d\cdot\log\pi_{k+1}(a|s;\boldsymbol{\lambda}) - \frac{1}{\alpha}\sup_{\boldsymbol{\lambda}'\in W_k(\boldsymbol{\lambda}),\boldsymbol{Q}'\in\mathcal{Q}_k}\boldsymbol{\lambda}^\top\boldsymbol{Q}'(s,a;\boldsymbol{\lambda}')\right]
$$

$$
\leq \mathbb{E}_{a\sim\pi_k(\cdot|s)}\left[\boldsymbol{\lambda}^\top\mathbf{1}_d\cdot\log\pi_k(a|s;\boldsymbol{\lambda}) - \frac{1}{\alpha}\sup_{\boldsymbol{\lambda}'\in W_k(\boldsymbol{\lambda}),\boldsymbol{Q}'\in\mathcal{Q}_k}\boldsymbol{\lambda}^\top\boldsymbol{Q}'(s,a;\boldsymbol{\lambda}')\right]. \quad (14)
$$

Next, we proceed to consider the multi-objective soft Bellman equation as follows:

$$\boldsymbol{\lambda}^\top\boldsymbol{Q}^{\pi_k}(s_0,a_0;\boldsymbol{\lambda}) - \boldsymbol{\lambda}^\top\boldsymbol{r}(s_0,a_0) \tag{15}$$

$$= \gamma\boldsymbol{\lambda}^\top\mathbb{E}_{s_1\sim P(\cdot|s_0,a_0),a_1\sim\pi_k(\cdot|s_1;\boldsymbol{\lambda})}\left[\boldsymbol{Q}^{\pi_k}(s_1,a_1;\boldsymbol{\lambda}) - \alpha\cdot\boldsymbol{\lambda}^\top\mathbf{1}_d\log(\pi_k(a_1|s_1;\boldsymbol{\lambda}))\right] \tag{16}$$

$$\leq \gamma\mathbb{E}_{s_1\sim P(\cdot|s_0,a_0),a_1\sim\pi_k(\cdot|s_1;\boldsymbol{\lambda})}\left[\sup_{\boldsymbol{\lambda}'\in W_k(\boldsymbol{\lambda}),\boldsymbol{Q}'\in\mathcal{Q}_k}\boldsymbol{\lambda}^\top\boldsymbol{Q}'(s_1,a_1;\boldsymbol{\lambda}') - \alpha\cdot\boldsymbol{\lambda}^\top\mathbf{1}_d\log(\pi_k(a_1|s_1;\boldsymbol{\lambda}))\right]$$

$$\tag{17}$$

$$\leq \gamma\mathbb{E}_{s_1\sim P(\cdot|s_0,a_0),a_1\sim\pi_{k+1}(\cdot|s_1;\boldsymbol{\lambda})}\left[\sup_{\boldsymbol{\lambda}'\in W_k(\boldsymbol{\lambda}),\boldsymbol{Q}'\in\mathcal{Q}_k}\boldsymbol{\lambda}^\top\boldsymbol{Q}'(s_1,a_1;\boldsymbol{\lambda}') - \alpha\cdot\boldsymbol{\lambda}^\top\mathbf{1}_d\log(\pi_{k+1}(a_1|s_1;\boldsymbol{\lambda}))\right]$$

$$\tag{18}$$

$$\leq -\gamma\mathbb{E}_{s_1\sim P(\cdot|s_0,a_0),a_1\sim\pi_{k+1}(\cdot|s_1;\boldsymbol{\lambda})}\left[\alpha\cdot\boldsymbol{\lambda}^\top\mathbf{1}_d\log(\pi_{k+1}(a_1|s_1;\boldsymbol{\lambda}))\right]$$

$$+ \gamma\mathbb{E}_{s_1\sim P(\cdot|s_0,a_0),a_1\sim\pi_{k+1}(\cdot|s_1;\boldsymbol{\lambda})}\left[\boldsymbol{\lambda}^\top\boldsymbol{r}(s_1,a_1) - \gamma\mathbb{E}_{s_2\sim P(\cdot|s_1,a_1),a_2\sim\pi_k(\cdot|s_2)}[\alpha\boldsymbol{\lambda}^\top\mathbf{1}_d\log(\pi_k(a_2|s_2;\boldsymbol{\lambda}))]\right.$$

$$\left. + \gamma\sup_{\boldsymbol{\lambda}'\in W_k(\boldsymbol{\lambda}),\boldsymbol{Q}'\in\mathcal{Q}_k}\mathbb{E}_{s_2\sim P(\cdot|s_1,a_1),a_2\sim\pi_k(\cdot|s_2)}\left[\boldsymbol{\lambda}^\top\boldsymbol{Q}'(s_2,a_2;\boldsymbol{\lambda})\right]\right] \tag{19}$$

$$\leq -\gamma\mathbb{E}_{s_1\sim P(\cdot|s_0,a_0),a_1\sim\pi_{k+1}(\cdot|s_1;\boldsymbol{\lambda})}\left[\alpha\cdot\boldsymbol{\lambda}^\top\mathbf{1}_d\log(\pi_{k+1}(a_1|s_1;\boldsymbol{\lambda}))\right]$$

$$+ \gamma\mathbb{E}_{s_1\sim P(\cdot|s_0,a_0),a_1\sim\pi_{k+1}(\cdot|s_1;\boldsymbol{\lambda})}\left[\boldsymbol{\lambda}^\top\boldsymbol{r}(s_1,a_1) - \gamma\mathbb{E}_{s_2\sim P(\cdot|s_1,a_1),a_2\sim\pi_{k+1}(\cdot|s_2)}[\alpha\boldsymbol{\lambda}^\top\mathbf{1}_d\log(\pi_{k+1}(a_2|s_2;\boldsymbol{\lambda}))]\right.$$

$$\left. + \gamma\sup_{\boldsymbol{\lambda}'\in W_k(\boldsymbol{\lambda}),\boldsymbol{Q}'\in\mathcal{Q}_k}\mathbb{E}_{s_2\sim P(\cdot|s_1,a_1),a_2\sim\pi_{k+1}(\cdot|s_2)}\left[\boldsymbol{\lambda}^\top\boldsymbol{Q}'(s_2,a_2;\boldsymbol{\lambda})\right]\right]$$

$$\tag{20}$$

$$\leq -\gamma \mathbb{E}_{s_1 \sim P(\cdot|s_0,a_0), a_1 \sim \pi_{k+1}(\cdot|s_1;\boldsymbol{\lambda})} \Big[ \alpha \cdot \boldsymbol{\lambda}^\top \mathbf{1}_d \log(\pi_{k+1}(a_1|s_1;\boldsymbol{\lambda})) \Big] \tag{21}$$

$$+ \mathbb{E}_{P,\pi_{k+1}} \Big[ \sum_{t \geq 1} \gamma^t \mathbb{E} \Big[ \boldsymbol{\lambda}^\top \boldsymbol{r}(s_t, a_t) \tag{22}$$

$$- \gamma \mathbb{E}_{s_{t+1} \sim P(\cdot|s_t,a_t), a_{t+1} \sim \pi_{k+1}(\cdot|s_{t+1})} \big[ \alpha \boldsymbol{\lambda}^\top \mathbf{1}_d \log(\pi_{k+1}(a_{t+1}|s_{t+1};\boldsymbol{\lambda})) | s_t, a_t \big] \Big] \Big] \tag{23}$$

$$= \boldsymbol{\lambda}^\top \boldsymbol{Q}^{\pi_{k+1}}(s_0, a_0; \boldsymbol{\lambda}) - \boldsymbol{\lambda}^\top \boldsymbol{r}(s_0, a_0), \tag{24}$$

where (16) follows from the multi-objective soft Bellman equation, (17) holds by the sup operation and the fact that $\boldsymbol{Q}^{\pi_k} \in \mathcal{Q}_k$, (18) follows from (14), (19) holds by applying the multi-objective soft Bellman equation to $\boldsymbol{Q}'(s_1, a_1; \boldsymbol{\lambda})$, (20) again follows from the inequality in (14), (23) is obtained by unrolling the whole trajectory, and (24) holds by the definition of $\boldsymbol{Q}^\pi$. $\qquad\square$

**Lemma 2 (Multi-Objective Soft Policy Evaluation)** *Under the multi-objective soft Bellman backup operator $\mathcal{T}_{MO}^\pi$ with respect to a policy $\pi$ and some $\boldsymbol{Q}^{(0)} : \mathcal{S} \times \mathcal{A} \to \mathbb{R}^d$, the sequence of intermediate $\boldsymbol{Q}$-functions $\{\boldsymbol{Q}^{(i)}\}$ during policy evaluation is given by $\boldsymbol{Q}^{(i+1)} = \mathcal{T}_{MO}^\pi \boldsymbol{Q}^{(i)}$, for all $i \in \mathbb{N} \cup \{0\}$. Then, $\boldsymbol{Q}^{(i)}$ converges to the soft $\boldsymbol{Q}$-function of $\pi$, as $i \to \infty$.*

**Proof (Lemma 2)** This can be directly obtained from the standard convergence property of iterative policy evaluation (Sutton and Barto, 2018) in two steps: (i) Define the entropy-augmented reward as $\boldsymbol{r}(s, a; \pi) := \boldsymbol{r}(s, a) + \gamma \mathbb{E}_{s' \sim \mathcal{P}(\cdot|s,a), a' \sim \pi(\cdot|s)}[\alpha \log \pi(a|s) \mathbf{1}_d]$, which is a bounded function. (ii) Then, rewrite the policy evaluation update as

$$\boldsymbol{Q}^\pi(s, a) \leftarrow \boldsymbol{r}(s, a; \pi) + \gamma \mathbb{E}_{s' \sim \mathcal{P}(\cdot|s,a), a' \sim \pi(\cdot|s)}[\boldsymbol{Q}^\pi(s', a'; \boldsymbol{\lambda})]. \tag{25}$$

This completes the proof. $\qquad\square$

Now we are ready to prove Theorem 3.1.

**Proof (Theorem 3.1)** Note that by Lemma 1, the sequence $\boldsymbol{\lambda}^\top \boldsymbol{Q}^{\pi_k}$ is monotonically increasing. As each element in $\boldsymbol{Q}^\pi$ is bounded above for all $\pi \in \Pi$ given the boundedness of both the reward and the entropy term, the sequence of policies shall converge to some policy $\pi^*$. The remaining thing is to show that $\pi^*$ is optimal: (i) Define $L_{\pi'}(\pi) := D_{\mathrm{KL}}\left(\pi(\cdot \mid s) \middle\| \frac{\exp\left(\sup_{\boldsymbol{\lambda} \in W(\boldsymbol{\lambda}), \boldsymbol{Q}' \in \mathcal{Q}} \boldsymbol{Q}'^\pi(s, \cdot; \boldsymbol{\lambda})\right)}{Z_\mathcal{Q}}\right)$.
(ii) Upon convergence, we shall have $L_{\pi^*}(\pi^*(\cdot|s)) \leq L_{\pi^*}(\pi(\cdot|s))$ for all $\pi \in \Pi$. Using the same iterative argument as in the proof of Lemma 1, we get $\boldsymbol{\lambda}^\top \boldsymbol{Q}^{\pi^*}(s, a; \boldsymbol{\lambda}) \geq \boldsymbol{\lambda}^\top \boldsymbol{Q}^\pi(s, a; \boldsymbol{\lambda})$ for all $(s, a) \in \mathcal{S} \times \mathcal{A}$ and all $\boldsymbol{\lambda} \in \Lambda$. $\qquad\square$

# B DETAILED CONFIGURATION OF EXPERIMENTS

## B.1 DETAILS ON THE EVALUATION DOMAINS

- Continuous Deep Sea Treasure (DST): DST is a classical multi-objective reinforcement learning environment. We control the agent to find the treasure, while the further the treasure is, the higher its value. In other words, the agent needs to spend more resources (-1 penalty for each action) to get the more precious treasure. To extend DST to continuous space, we modify the simple four direction movement to the movement in a circle, we set the $\beta$ of DST to 3.

- Multi-Objective Continuous LunarLander: We modify LunarLander to the multi-objective version by dismantling the reward to main engine cost, side engine cost, shaping reward, and result reward. Since the past MORL methods were conducted in environments with 2 or 3 objectives, we created an environment with 4 and 5 objectives to show our method can be used in high dimension objectives environments, we set the $\beta$ of LunarLander to 5.

- MuJoCo: We divide the scalar reward in MuJoCo environments into vector rewards. What's more, we amplify the weight of the control cost to make the magnitude of each reward element similar.

    - **HalfCheetah2d:** 2 objectives as forward speed, control cost ($\mathcal{S} \subseteq \mathbb{R}^{17}, \mathcal{A} \subseteq \mathbb{R}^6$), 1000 times for control cost, and $\beta = 10$.

- **Hopper2d:** 2 objectives: forward speed, control cost ($\mathcal{S} \subseteq \mathbb{R}^{11}, \mathcal{A} \subseteq \mathbb{R}^3$), 1500 times for control cost, and $\beta = 3$.
- **Hopper3d:** 3 objectives: forward speed, jump reward, control cost ($\mathcal{S} \subseteq \mathbb{R}^{11}, \mathcal{A} \subseteq \mathbb{R}^3$), 1500 times for control cost. The jump reward is 15 times of the difference between current height and initial height, and $\beta = 10$.
- **Hopper5d:** 5 objectives: forward speed, control cost of each of the 3 joints, and healthy reward ($\mathcal{S} \subseteq \mathbb{R}^{11}, \mathcal{A} \subseteq \mathbb{R}^3$), 1500 times for control cost, and $\beta = 5$
- **Ant2d:** 2 objectives: forward speed, control cost ($\mathcal{S} \subseteq \mathbb{R}^{111}, \mathcal{A} \subseteq \mathbb{R}^8$), 1 times for control cost, and $\beta = 10$.
- **Ant3d:** 3 objectives: forward speed, control cost, healthy reward ($\mathcal{S} \subseteq \mathbb{R}^{111}, \mathcal{A} \subseteq \mathbb{R}^8$), 1 times for control cost, 1 times for healthy reward, and $\beta = 10$.
- **Walker2d:** 2 objectives: forward speed, control cost ($\mathcal{S} \subseteq \mathbb{R}^{17}, \mathcal{A} \subseteq \mathbb{R}^6$), 1000 times for control cost, and $\beta = 3$.

## B.2 HYPERPARAMETERS

### B.2.1 HYPERPARAMETERS OF Q-PENSIEVE

We conduct all experiment on baselines with following hyperparameters.

Table 2: Hyperparameters of $Q$-Pensieve.

| Parameter | Value |
|---|---|
| Optimizer | Adam |
| Learning Rate | 0.0003 |
| Discount Factor | 0.99 |
| Replay Buffer Size | 1000000 |
| Depth of Hidden Layers | 2 |
| Number of Hidden Units per Layer | 256 |
| Number of Samples per Minibatch | 256 |
| Nonlinearity | ReLU |
| Target Smoothing Coefficient | 0.005 |
| Target Update Interval | 1 |
| Gradient Steps | 1 |

### B.2.2 HYPERPARAMETERS OF PGMORL AND PFA

For PGMORL and PFA, we use the hyperparameters as provided in Table 3:

- n: the number of reinforcement learning tasks.
- total_steps: the total number of environment training steps.
- $m_w$: the number of iterations in warm-up stages.
- $m_t$: the number of iterations in evolutionary stages.
- $P_{\text{num}}$: the number of performance buffers.
- $P_{\text{size}}$: the size of each performance buffer.
- $n_{\text{weight}}$: the number of sampled weights for each policy.
- sparsity: the weight of sparsity metric.

## C PSEUDO CODE OF Q-PENSIEVE

We provide the pseudo code in Algorithm 1 as follows.

Table 3: Hyperparameters of PGMORL and PFA.

| Environments | $n$ | total_steps | $m_w$ | $m_t$ | $P_{\text{num}}$ | $P_{\text{size}}$ | $n_{\text{weight}}$ | sparsity |
|---|---|---|---|---|---|---|---|---|
| DST2d | 5 | $1.5 \times 10^6$ | 80 | 20 | 100 | 2 | 7 | $-1$ |
| LunarLander4d | 35 | $7.5 \times 10^6$ | 40 | 10 | 400 | 2 | 7 | $-10^6$ |
| LunarLander5d | 35 | $7.5 \times 10^6$ | 40 | 10 | 400 | 2 | 7 | $-10^6$ |
| HalfCheetah2d | 5 | $1.5 \times 10^7$ | 80 | 20 | 100 | 2 | 7 | $-1$ |
| Hopper2d | 5 | $4.5 \times 10^6$ | 200 | 40 | 100 | 2 | 7 | $-1$ |
| Hopper3d | 15 | $1.5 \times 10^7$ | 200 | 40 | 200 | 2 | 7 | $-10^6$ |
| Hopper5d | 35 | $7.5 \times 10^6$ | 200 | 40 | 400 | 2 | 7 | $-10^6$ |
| Ant2d | 5 | $1.5 \times 10^7$ | 200 | 40 | 100 | 2 | 7 | $-1$ |
| Ant3d | 15 | $1.5 \times 10^7$ | 200 | 40 | 200 | 2 | 7 | $-10^6$ |
| Walker2d | 5 | $4.5 \times 10^6$ | 80 | 20 | 100 | 2 | 7 | $-1$ |

---

**Algorithm 1:** $Q$-Pensieve

**Input** : $\phi_1, \phi_2, \theta$, preference sampling distribution $\mathcal{P}_{\boldsymbol{\lambda}}$, number of preference vectors $N_{\boldsymbol{\lambda}}$, the soft update coefficient $\tau$, actor learning rates $\eta_\pi$, critic learning rates $\eta_Q$

**Output** : $\phi_1, \phi_2, \theta$

1   $\bar{\phi}_1 \leftarrow \phi_1, \bar{\phi}_2 \leftarrow \phi_2$;
2   $\mathcal{M} \leftarrow \emptyset$         ▷ Initialize replay buffer;
3   $\mathcal{B} \leftarrow \emptyset$;
4 **for** *each iteration* **do**
5    *sample weight $\boldsymbol{\lambda}$ from $\Lambda$ according $\mathcal{P}_{\boldsymbol{\lambda}}$* ;
6    **for** *each environment step* **do**
7      $a_t \sim \pi_\theta(\cdot|s_t; \boldsymbol{\lambda})$;
8      $s_{t+1} \sim P(\cdot|s_t, a_t)$;
9      $\mathcal{M} \leftarrow \mathcal{M} \bigcup \{(s_t, a_t, r(s_t, a_t), s_{t+1})\}$;
10    **for** *each gradient step* **do**
11      *sample $N_{\boldsymbol{\lambda}} - 1$ preferences and add them to set $W$*;
12      $W \leftarrow W \bigcup \{\boldsymbol{\lambda}\}$;
13      $\phi_i \leftarrow \phi_i - \eta_Q \hat{\nabla}_{\phi_i} \mathcal{L}_Q(\phi_i; \boldsymbol{\lambda}), \mathcal{B} \leftarrow \mathcal{B} \bigcup \{\phi_i; \boldsymbol{\lambda}\}$ for $i \in \{1, 2\}$;
14      *compute $\hat{\nabla}_\theta$ with eq. 11 with $W$*;
15      $\theta \leftarrow \theta - \eta_\pi \hat{\nabla}_\theta \mathcal{L}_\pi(\theta; \boldsymbol{\lambda})$;
16      $\bar{\phi}_i \leftarrow \tau \phi_i + (1 - \tau) \bar{\phi}_i$ for $i \in \{1, 2\}$;

---

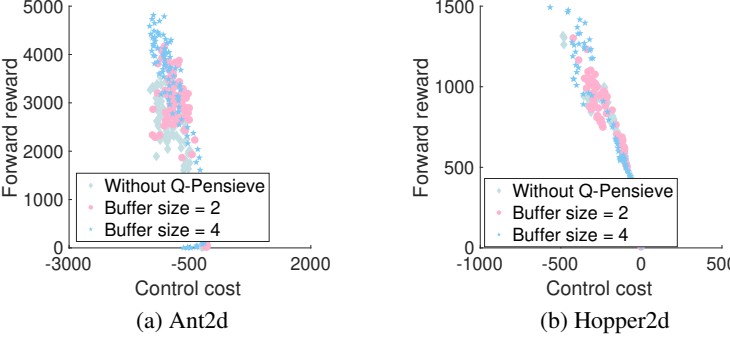

(a) Ant2d          (b) Hopper2d

Figure 6: Return vectors attained under different $Q$ replay buffer sizes of Q-Pensieve

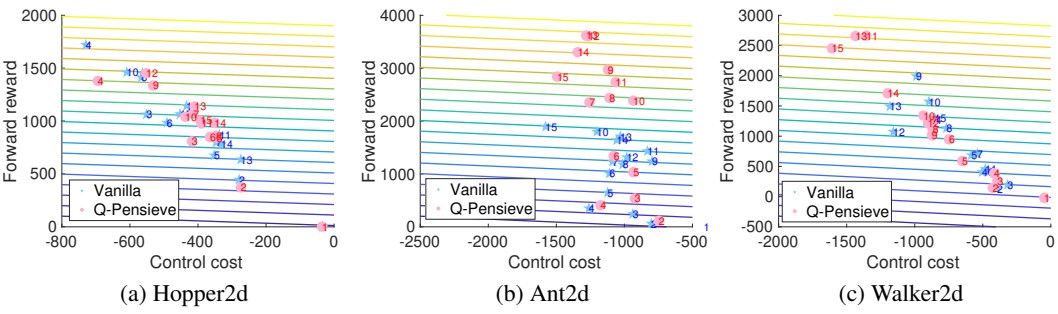

Figure 7: Return vectors attained under preference $\boldsymbol{\lambda} = [0.9, 0.1]$ at different training stages. A number $x$ on the red or blue marker indicates that the model is obtained at $10 \cdot x$ million steps.

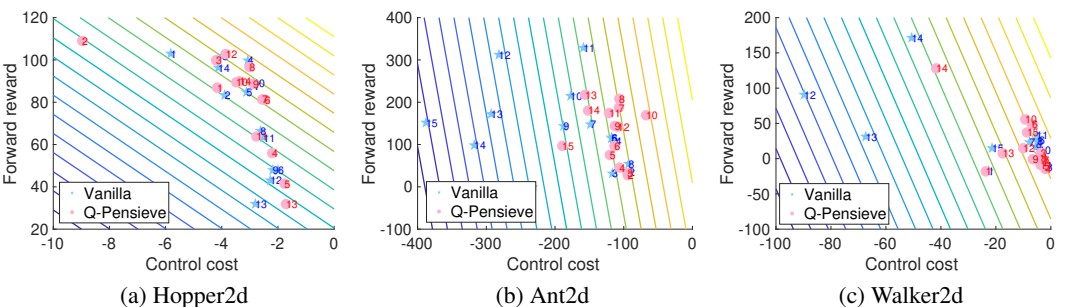

Figure 8: Return vectors attained under preference $\boldsymbol{\lambda} = [0.1, 0.9]$ at different training stages. A number $x$ on the red or blue marker indicates that the model is obtained at $10 \cdot x$ million steps.

# D COMPARISON OF LEARNING CURVES

We demonstrate the learning curves of $\boldsymbol{Q}$-Pensieve and the benchmark methods. In Figures 9-15, we can find that $\boldsymbol{Q}$-Pensieve enjoys the fastest learning progress in almost all tasks and preferences. Notably, as PGMORL is an evolutionary method and does explicit search for policies for only a small set of preferences vectors in each generation, the typical learning curve (in terms of expected total reward) under a given preference is not very informative about the overall learning progress. Therefore, regarding the learning curves, we compare $\boldsymbol{Q}$-Pensieve with CN-DER and PFA. Moreover, as PFA cannot handle tasks with more than two objectives (this fact has also been mentioned in (Xu et al., 2020)), PFA is evaluated only in tasks with two objectives.

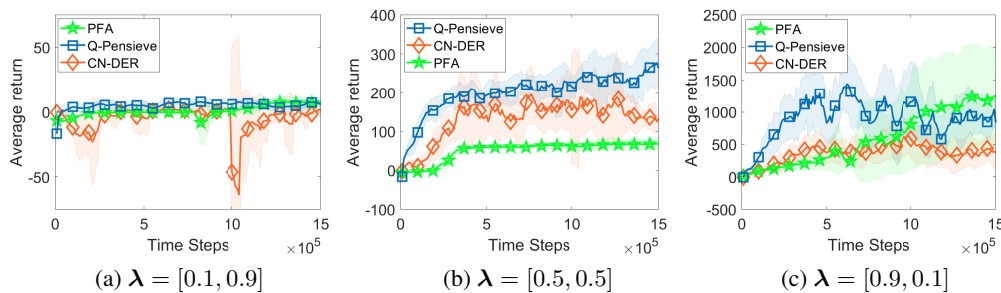

Figure 9: Average return in Hopper2d over 5 random seeds (average return is the inner product of the reward vectors and the corresponding preference).

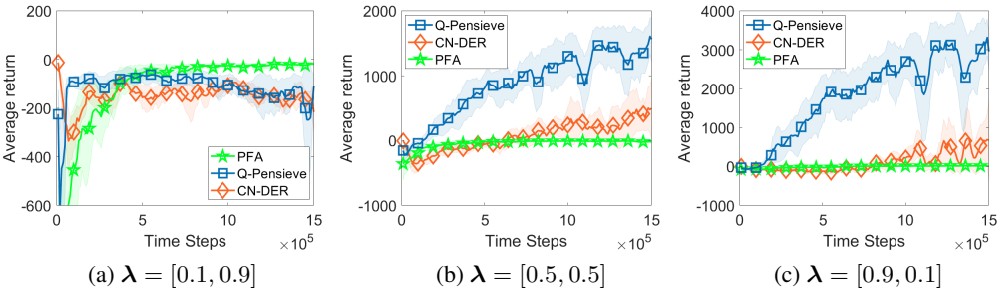

(a) $\boldsymbol{\lambda} = [0.1, 0.9]$ (b) $\boldsymbol{\lambda} = [0.5, 0.5]$ (c) $\boldsymbol{\lambda} = [0.9, 0.1]$

Figure 10: Average return in Ant2d over 5 random seeds (average return is the inner product of the reward vectors and the corresponding preference).

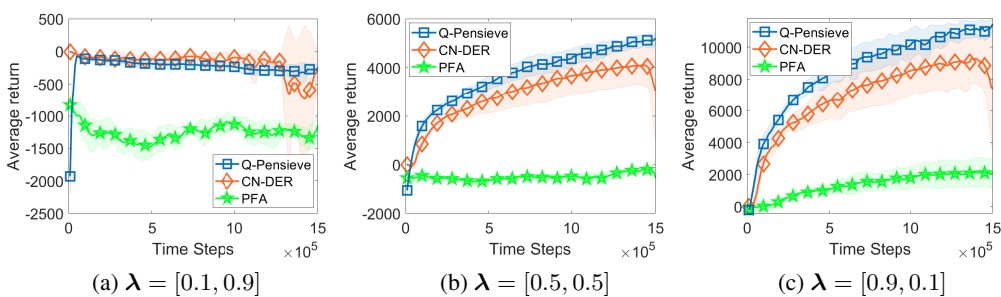

(a) $\boldsymbol{\lambda} = [0.1, 0.9]$ (b) $\boldsymbol{\lambda} = [0.5, 0.5]$ (c) $\boldsymbol{\lambda} = [0.9, 0.1]$

Figure 11: Average return in HalfCheetah2d over 5 random seeds (average return is the inner product of the reward vectors and the corresponding preference).

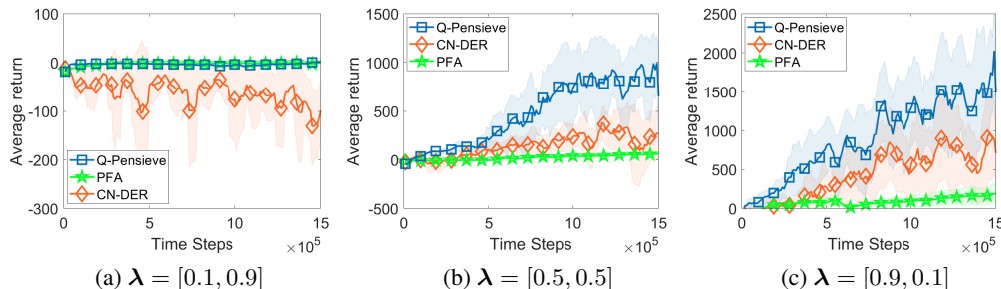

(a) $\boldsymbol{\lambda} = [0.1, 0.9]$ (b) $\boldsymbol{\lambda} = [0.5, 0.5]$ (c) $\boldsymbol{\lambda} = [0.9, 0.1]$

Figure 12: Average return in Walker2d over 5 random seed (average return is the inner product of the reward vectors and the corresponding preference).

## E  ADDITIONAL EXPERIMENTAL RESULTS

In this section, we compare $\boldsymbol{Q}$-Pensieve with the baseline methods, discuss how the performance of $\boldsymbol{Q}$-Pensieve can be further improved through hyperparameter tuning, and then demonstrate the model generalization of $\boldsymbol{Q}$-Pensieve.

### E.1  COMPARISON WITH THE ENVELOPE Q-LEARNING ALGORITHM

The Envelope $\boldsymbol{Q}$-Learning algorithm and its neural version Envelope DQN (Yang et al., 2019) presume that the action space is discrete. To adapt Envelope DQN to the continuous control tasks

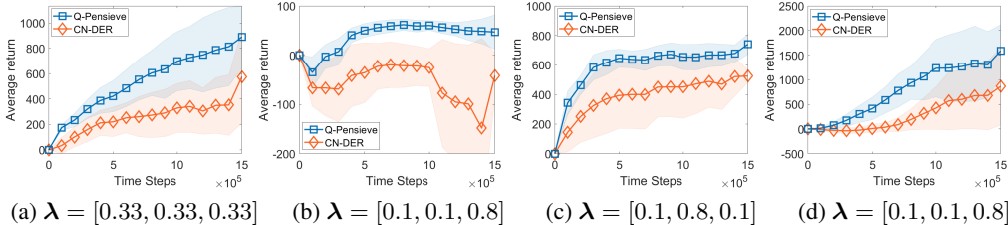

Figure 13: Average return in Ant3d over 5 random seeds (average return is the inner product of the reward vectors and the corresponding preference).

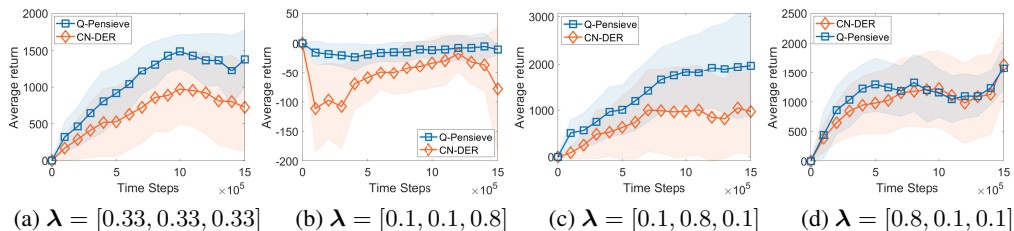

Figure 14: Average return in Hopper3d over 5 random seeds (average return is the inner product of the reward vectors and the corresponding preference).

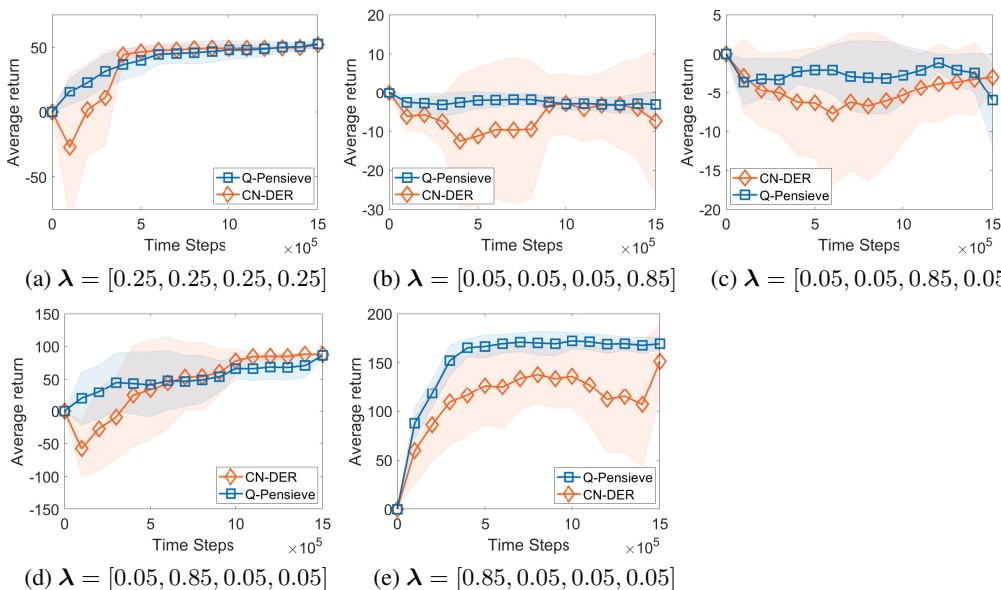

Figure 15: Average return in LunarLander4d over 5 random seeds (average return is the inner product of the reward vectors and the corresponding preference).

considered in our paper (including MuJoCo and continuous Deep Sea Treasure), we take the open-source implementation in (Yang et al., 2019) and apply action discretization, which has been shown to be also quite effective in MuJoCo control tasks (Tavakoli et al., 2018; Tang and Agrawal, 2020). We compare Envelope DQN with $Q$-Pensieve in both Hopper3d and DST2d environments. For Envelope DQN, we set the number of bins for each action dimension to be 11 and 5 for DST2d (actions are 2-dimensional) and Hopper3d (actions are 3-dimensional) respectively, based on the suggestions

provided by (Tavakoli et al., 2018). Table 4 shows the performance of $Q$-Pensieve and Envelope DQN in terms of the two metrics. We can observe that $Q$-Pensieve outperforms Envelope DQN by a large margin in the above two popular multi-objective tasks.

Table 4: Comparison of $Q$-Pensieve and Envelope DQN in terms of the two metrics across two domains. We report the mean and standard deviation over five random seeds.

| Environments | Metrics | Envelope DQN | $Q$-Pensieve |
|---|---|---|---|
| DST2d | HV($\times 10^2$) | $7.02_{\pm 0.24}$ | $\mathbf{10.21_{\pm 1.40}}$ |
| | UT($\times 10^0$) | $4.61_{\pm 0.08}$ | $\mathbf{7.31_{\pm 0.91}}$ |
| Hopper3d | HV($\times 10^5$) | $0.43_{\pm 0.21}$ | $\mathbf{13.31_{\pm 2.03}}$ |
| | UT($\times 10^2$) | $-0.39_{\pm 0.26}$ | $\mathbf{4.08_{\pm 1.10}}$ |

### E.2 EMPIRICAL STUDY OF Q-PENSIEVE

**$Q$ Replay Buffer Size:** One could expect that a larger $Q$ buffer size could help provide a more diverse collection of $Q$ snapshots and thereby better boost the policy improvement update in each iteration. On the other hand, in practice, the required memory usage also scales with the $Q$ buffer size. We evaluate $Q$-Pensieve under buffer sizes = 2, 4, 6 and compare it to that without using a $Q$ replay buffer in Ant2d. Table 5 show that empirically a relatively small $Q$ buffer size already offers a significant performance improvement.

**$Q$ Replay Buffer Update Interval:** To ensure that the $Q$ snapshots in the $Q$ replay buffer are rather diverse, we would suggest that the update interval shall not be too small (otherwise the Q snapshots in the buffer would be fairly similar). Moreover, as in general this update interval can be viewed as a hyperparameter to be tuned (similar to the update interval of the target networks in many RL algorithms). We further do an empirical study on the performance of $Q$-Pensieve under different update intervals. We run $Q$-Pensieve with different update intervals in Ant2d for 1500k steps. Table 6 show that the hypervolume is not sensitive to the update interval, and the performance in UT can potentially be further improved through hyperparameter tuning.

### E.3 MODEL GENERALIZATION OF Q-PENSIEVE

To demonstrate that the critic model with the preference vector can generalize well, we define a metric for the critic as

$$\mathcal{L}_{\text{critic}} = \mathbb{E}_{s \sim \mathcal{S}, a \sim \mathcal{A}} || \boldsymbol{Q}(s, a; \boldsymbol{\lambda}) - \boldsymbol{Q}_{\text{true}}(s, a; \boldsymbol{\lambda}) ||_2, \quad (26)$$

where $\boldsymbol{Q}$ is the action-value function learned by our critic network and $\boldsymbol{Q}_{\text{true}}$ is the true $Q$ function calculated by Monte-Carlo method. Table 7 and Table 8 show the $\mathcal{L}_{\text{critic}}$ under various preferences $\boldsymbol{\lambda}$ at different training stages in Hopper2d and HalfCheetah2d respectively. Note that the true $Q$ values are typically in the range of 1000 to a few thousands. Therefore, we can see that $\mathcal{L}_{\text{critic}}$ under various preferences is indeed pretty low, which indicates that the critic model can generalize well across preferences.

Table 5: Comparison of $Q$-Pensieve with different $Q$ replay buffer size in terms of the two metrics in Ant2d over five random seeds.

| Metrics | **Without Q Buffer** | **Size = 2** | **Size = 4** | **Size = 6** |
|---|---|---|---|---|
| HV($\times 10^7$) | $0.93_{\pm 0.25}$ | $0.99_{\pm 0.23}$ | $1.00_{\pm 0.18}$ | $1.27_{\pm 0.14}$ |
| UT($\times 10^2$) | $8.17_{\pm 4.83}$ | $12.45_{\pm 2.75}$ | $14.04_{\pm 3.03}$ | $15.31_{\pm 3.47}$ |

Table 6: Comparison of $Q$-Pensieve with different replay buffer update interval in terms of the two metrics in Ant2d over five random seeds.

| Metrics | Interval = 500 | Interval = 1000 | Interval = 1500 | Interval = 2000 |
|---|---|---|---|---|
| HV($\times 10^6$) | $8.30_{\pm 0.60}$ | $9.90_{\pm 2.31}$ | $8.63_{\pm 1.31}$ | $8.53_{\pm 1.63}$ |
| UT($\times 10^2$) | $8.94_{\pm 2.01}$ | $12.45_{\pm 2.75}$ | $11.83_{\pm 1.42}$ | $13.33_{\pm 2.97}$ |

Table 7: $\mathcal{L}_{\text{critic}}$ in Hopper2d over five random seeds.

| Preferences | 100K steps | 500K steps | 1000K steps | 1500K steps |
|---|---|---|---|---|
| $\lambda$=[0.1,0.9] | 23.74 | 43.59 | 38.96 | 41.47 |
| $\lambda$=[0.3,0.7] | 22.21 | 14.82 | 15.10 | 14.65 |
| $\lambda$=[0.5,0.5] | 27.59 | 14.45 | 24.91 | 11.59 |
| $\lambda$=[0.7,0.3] | 47.54 | 10.25 | 15.17 | 11.52 |
| $\lambda$=[0.9,0.1] | 87.01 | 46.95 | 84.76 | 31.96 |

Table 8: $\mathcal{L}_{\text{critic}}$ in HalfCheetah2d over five random seeds.

| Preferences | 100K steps | 500K steps | 1000K steps | 1500K steps |
|---|---|---|---|---|
| $\lambda$=[0.1,0.9] | 62.92 | 139.39 | 118.135 | 92.35 |
| $\lambda$=[0.3,0.7] | 134.1 | 174.81 | 132.305 | 131.71 |
| $\lambda$=[0.5,0.5] | 192.47 | 113.20 | 98.18 | 90.07 |
| $\lambda$=[0.7,0.3] | 169.70 | 87.87 | 83.21 | 74.69 |
| $\lambda$=[0.9,0.1] | 146.71 | 66.92 | 65.09 | 63.85 |

