# OpenReview forum: "Q-Pensieve: Boosting Sample Efficiency of Multi-Objective RL Through Memory Sharing of Q-Snapshots"
_ICLR.cc/2023/Conference — ICLR 2023 poster_

### Official Review · Reviewer_VKcR · 2022-10-24

**Confidence:** 4
**Correctness:** 4
**Technical Novelty And Significance:** 3
**Empirical Novelty And Significance:** 3
**Recommendation:** 8

**Clarity, Quality, Novelty And Reproducibility:**

This paper is generally clear and well written. One important sentence that needs to be fixed is that in the ‘Implicit search’ part of the Related Work- ‘While there is implcit’. This sentence is clearly incomplete.

There were also a few minor typos:
Equations 1 and 5 should have V(s’) instead of V(s).
Page 8 has ‘Q reply buffer’ instead of ‘Q replay buffer’


**Strength And Weaknesses:**

The ideas presented in this work are generally well presented and the various sections do a good job of introducing the various background concepts. The method itself is quite easy to understand although I have not looked into the proof in the Appendix closely.

While I generally think the paper is well written, there are some questions that I had regarding the implementation and results. One point in particular that could be improved is that the authors should mention earlier in the text the expected size of the replay buffer being used for the Q snapshots. It was not until the experiments that I saw the replay size of 4 was mentioned. On first reading it was not clear how big the replay size was and since this can have a large memory footprint the size matters quite a bit.

In the experimental section episode dominance is defined as a metric that directly compares a pair of algorithms. In Table 1 this metric is presented for all the baseline methods - how is it computed here? My understanding is that this metric defines on average for how many preferences does algorithm 1 out-perform algorithm 2. But it isn’t clear what the two algorithms chosen in practice are.

The idea of using different Q networks for learning has also been studied in single objective RL. For example, TD-3 often outperforms SAC in single-objective RL using 2 Q networks in the update rule. There are differences in the two approaches but I am curious how well a TD-3 like update (extended to the multi-objective setting) compares to the method being proposed in the paper? Even with a Q buffer size of 4, the memory footprint of the proposed approach is quite large so it would be compelling if a simpler method worked as well.

Additionally the mechanism of information sharing through the Q replay is not clear to me. How is the replay buffer implemented? With such a small size is it effectively a queue containing the last N snapshots of learning? What happens when the replay is updated every N/2nd iteration and only stores N/2 snapshots instead say? As things stand I think the idea has merit but I think more can be done to understand exactly where the benefits are coming from which could perhaps simplify the method.

EDIT (Post author discussion period):

I would like to thank the authors for responding to my feedback and updating the paper with my comments. After looking through the discussion between the authors and the other reviewers and the latest draft of the paper, I think this is in good shape to be accepted.


**Summary Of The Paper:**

This paper presents a method, dubbed Q-Pensieve, for reusing critic knowledge across training iterations in a multi-objective reinforcement learning (MORL) setting. Specifically it proposes a modification of Soft Actor Critic (SAC) in the multi-objective setting that uses snapshots of previous Q functions in the learning update for which a convergence proof is provided. Finally the proposed algorithm is evaluated on a number of continuous control domains against other benchmarks and across 3 metrics and some ablations are considered.

**Summary Of The Review:**

Overall I think this paper is generally well presented and meets the acceptance bar. I am happy to increase my score once my clarification questions and results are presented.

---

> ### Author Response · Authors · 2022-11-16
> **Response to Reviewer VKcR**
>
> **(D3) The idea of using different Q networks for learning has also been studied in single objective RL. For example, TD-3 often outperforms SAC in single-objective RL using 2 Q networks in the update rule. There are differences in the two approaches but I am curious how well a TD-3 like update (extended to the multi-objective setting) compares to the method being proposed in the paper? Even with a Q buffer size of 4, the memory footprint of the proposed approach is quite large so it would be compelling if a simpler method worked as well.**
>
> Thank you for the question. We would like to highlight that Q-Pensieve and the twin Q networks of TD3 are two orthogonal techniques and could complement each other in MORL. Specifically, Q-Pensieve is meant to boost the multi-objective *policy improvement* while the twin Q networks are mainly to circumvent the overestimation issue in *policy evaluation*.
>
> In terms of implementation, our implementation of Q-Pensieve follows a similar design as the SAC of OpenAI Spinning Up framwork (https://spinningup.openai.com/en/latest/), which already incorporates the TD3-type critic into the policy evaluation step of SAC.
> Moreover, we do observe that Q-Pensieve could benefit from the TD3-type critic as the Q-Pensieve policy improvement step could be more effective with more accurate Q estimates.
>
> Regarding the memory usage, as described in our response (All-3), in the Ant task, each Q snapshot takes about 15MB of GPU memory in our experiments, and this appears quite acceptable in practice.
>
>
> **(D4) Additionally the mechanism of information sharing through the Q replay is not clear to me. How is the replay buffer implemented? With such a small size is it effectively a queue containing the last N snapshots of learning? What happens when the replay is updated every N/2nd iteration and only stores N/2 snapshots instead say? As things stand I think the idea has merit but I think more can be done to understand exactly where the benefits are coming from which could perhaps simplify the method.**
>
> Thank you for pointing this out. Yes, our Q replay buffer is implemented as a FIFO. For more details about the implementation of Q buffer, please see our response (All-3) above.
> Regarding the update interval, we would like to highlight that the update interval of Q replay buffer has negligible effect on the GPU memory usage and computation time as the pop/push operations of a FIFO only takes minimal computation time and these operations do not take extra memory. In all our experiments, we set the update interval to be 1000 steps without any tuning.
>
> Nevertheless, as this update interval can be viewed as a hyperparameter (similar to the update interval of the target networks in many RL algorithms), we further do an empirical study on the performance of Q-Pensieve under different update intervals. Specifically, we run Q-Pensieve with different update intervals in Ant for 700k steps (with Q buffer size = 2, as suggested by the reviewer). The experimental results are provided in the table below:
>
> | Update Interval (steps) | 500 | 1000 | 1500 | 2000 |
> | -------- | -------- | -------- |-------- | -------- |
> | HV($\times 10^{6}$)     | 6.59    | 7.47    |6.89     | 6.87     |
> | UT($\times 10^{2}$)    | 2.64     | 3.00     |2.94     | 4.90|
>
> We can observe that the hypervolume is not sensitive to the update interval, and the performance in UT can potentially be further improved through hyperparameter tuning.
>
> **(D5) There were also a few minor typos: Equations 1 and 5 should have V(s’) instead of V(s). Page 8 has ‘Q reply buffer’ instead of ‘Q replay buffer’.**
>
> Thank you for catching this. We have fixed the typos.

---

> > ### Author Response · Authors · 2022-12-02
> > **Response to Reviewer VKcR**
> >
> > Thank you for your constructive feedback and for editing the review comments. We are glad to see that the reviewer appreciate our contributions in presenting the Q-Pensieve method for MORL. If you have any further suggestions, please let us know. We will be more than happy to address them as well.

---

> ### Author Response · Authors · 2022-11-16
> **Response to Reviewer VKcR**
>
> We sincerely thank the reviewer for the positive feedback and the helpful suggestions. We provide our response as follows.
>
> **(D1) One point in particular that could be improved is that the authors should mention earlier in the text the expected size of the replay buffer being used for the Q snapshots. It was not until the experiments that I saw the replay size of 4 was mentioned. On first reading it was not clear how big the replay size was and since this can have a large memory footprint the size matters quite a bit.**
>
> Thank you for the suggestion. We agree with the reviewer that the expected size of the Q replay buffer could be mentioned earlier (and this is one strength of Q-Pensieve given that the size could be quite small). Accordingly, we have added one sentnece to highlight this fact at the end of Section 3.
> Moreover, as described in (All-2) of Response to All Reviewers, we provide an empirical study on the performance versus Q buffer size and show that Q-Pensieve can indeed achieve quite significant performance improvement with a relatively small Q buffer size.
>
>
> **(D2) In the experimental section episode dominance is defined as a metric that directly compares a pair of algorithms. In Table 1 this metric is presented for all the baseline methods - how is it computed here? My understanding is that this metric defines on average for how many preferences does algorithm 1 out-perform algorithm 2. But it isn’t clear what the two algorithms chosen in practice are.**
>
> Thank you for the question. As suggested by the reviewer, ED is meant to do a pariwise comparison of two algorithms in a preference-by-preference manner by measuring the percentage of preference vector under which  algorithm 1 can outperform algorithm 2. To ensure a fair comparison, for all the benchmark methods and Q-Pensieve, we choose vanilla MOSAC (i.e., a multi-objective extension of SAC without using Q-Pensieve) as our algorithm 2 for ED.

---

### Official Review · Reviewer_g5HG · 2022-10-24

**Confidence:** 3
**Clarity, Quality, Novelty And Reproducibility:** 1.The paper is well written.

2.The p…
**Correctness:** 3
**Technical Novelty And Significance:** 2
**Empirical Novelty And Significance:** 3
**Recommendation:** 5

**Strength And Weaknesses:**

Strength:

1.The paper study an important problem and the paper is well written.

2.The performance improvement is significant.

3.Theoretical analysis is interesting.

Weakness:

1.The idea that uses replay buffer is not quite novel. It is common for rl methods trained with replay buffer. I think it is novel to use the learned critic in previous rounds, can this technique helps for single RL algorithms?

2.The performance curve of each algorithm during the training is missing.

3.For the experimental details, how is the preference vector decided in practice?

4.The critic function is related with the preference vector, can does the model generalize well on this variable?

5.How the performance change when the number of objectives increases?

**Summary Of The Paper:**

This paper study the problem of  multiple objective reinforcement learning. As previous methods that search for Pareto front are not sample efficient. The paper propose the Q-Pensieve method, which updates the policy with learned Q-networks from past iterations. The paper provide some theoretical analysis. Finally, experiments on several multi-objectives environments validate its effectiveness over stoa baselines.

**Summary Of The Review:**

Overall, the paper is an interesting. A new MORL method is proposed to exploit the learned critic in previous rounds. The performance improvement is significant. But some detailed analysis and the discussion of the limitation are missing.

---

> ### Author Response · Authors · 2022-11-16
> **Response to Reviewer g5HG**
>
> **(C2) The performance curve of each algorithm during the training is missing.**
> Thank you for the suggestion. We have added the learning curves in Appendix D of the updated manuscript.
> Notably, as PGMORL is an evolutionary method and does explicit search for policies for only a small set of preferences vectors in each generation, the typical learning curve (in terms of expected total reward) under a given preference is not very informative about the overall learning progress. Therefore, regarding the learning curves, we compare Q-Pensieve with CN-DER and PFA. Moreover, as PFA cannot handle tasks with more than two objectives (this fact has also been mentioned in [Xu et al., 2020]), PFA is evaluated only in tasks with two objectives.
>
>
> **(C3) The critic function is related with the preference vector. Can the model generalize well on this variable?**
> To demonstrate that the critic model with the preference vector can generalize well, we define a metric for the critic as $L_{Critic}(\lambda)$ = $E_{(s,a)\sim D}||(Q(s,a;\lambda)-Q_{true}(s,a;\lambda)||\_2$, where $Q$ is the action-value function learned by our critic network and $Q_{true}$ is the true Q function caculated by Monte-Carlo method. We show the $L_{Critic}$ under various preferences $\lambda$ at different training stages in Hopper and HalfCheetah averaged over 5 seeds.
> $L_{Critic}$ in Hopper:
> |  | 100K steps | 500K steps |1000K steps | 1500K steps |
> | -------- | -------- | -------- |-------- | -------- |
> | $\lambda=[0.1, 0.9]$     | 23.74     | 43.59     |38.96     | 41.47     |
> | $\lambda=[0.3, 0.7]$     | 22.21     | 14.82     |15.10     | 14.65     |
> | $\lambda=[0.5, 0.5]$     | 27.59     | 14.45     |24.91     | 11.59     |
> | $\lambda=[0.7, 0.3]$     | 47.54     | 10.25     |15.17     | 11.52     |
> | $\lambda=[0.9, 0.1]$     | 87.01     | 46.95     |84.76     | 31.96     |
>
> $L_{Critic}$ in HalfCheetah:
> |  | 100K steps | 500K steps |1000K steps | 1500K steps |
> | -------- | -------- | -------- |-------- | -------- |
> | $\lambda=[0.1, 0.9]$     | 62.92     | 139.39     |118.135     | 92.35     |
> | $\lambda=[0.3, 0.7]$     | 134.1     | 174.81     |132.305     | 131.71     |
> | $\lambda=[0.5, 0.5]$     | 192.47     | 113.20     |98.18     | 90.07     |
> | $\lambda=[0.7, 0.3]$     | 169.70     | 87.87     |83.21     | 74.69     |
> | $\lambda=[0.9, 0.1]$     | 146.71     | 66.92     |65.09     | 63.85     |
>
> Note that in these two two-objective tasks, the true Q values are typically in the range of 1000 to a few thousands. Therefore, we can see that $L_{Critic}$ under various preferences is indeed pretty low, which indicates that the critic model can generalize well across preferences.
>
> **(C4) How the performance change when the number of objectives increases?**
> As shown in Table 1 of the paper, Q-Pensieve consistently outperforms the benchmark methods in tasks up to 4 objectives. To further demonstrate the ability of Q-Pensieve to handle more objectives, we also evaluate Q-Pensieve in tasks with 5 objectives (namely Hopper5d and LunarLander5d) and show that Q-Pensieve remains strong compared to the benchmark MORL methods. Please see (All-1) of the Response to All Reviewers for more details.
>
> **(C5) For the experimental details, how is the preference vector decided in practice?**
> To ensure that the agent can learn well-performing policies for various preferences, in each iteration we draw one preference vector (denoted by $\lambda$) randomly from some distribution $\mathcal{D}$, and the actor would be updated with respect to this $\lambda$. In our experiments, for tasks with $d$ objectives, we simply generate $d$ uniform random variables between zero and one and apply normalization such that the sum is one. If one would like to optimize this part of Q-Pensieve, one possible way is to leverage the Dirichlet distribution as done in CN-DER.

---

> ### Author Response · Authors · 2022-11-16
> **Response to Reviewer g5HG**
>
> **(C1) "I think it is novel to use the learned critic in previous rounds, can this technique helps for single RL algorithms?"**
>
> Thank you for appreciating the novelty and bringing up this insightful question. Yes, the technique of Q-Pensieve and Q replay buffer can also benefit single-objective RL. Specifically, we highlight several potential use cases as follows:
>
> * (i) Single-objective RL with exploration bonus or reward shaping:
> Various existing single-objective RL algorithms encourage exploration by incorporating an additional reward bonus term for encouraging exploration or curiosity (such as [Houthooft et al., 2016; Kim et al., 2019; Badia et al., 2020]. In these methods for single-objective RL, the weight of this additional bonus/penalty is typically viewed as a hyperparameter to be tuned. By contrast, we can instead view these problems as multi-objective RL problems, where the reward is a vector with the two entries being extrinsic reward and intrinsic reward. From this perspective, one can leverage Q-Pensieve to learn a collection of policies (one for each preference over the extrinsic reward and the exploration bonus) during training and thereafter select a policy that achieves high total extrinsic reward.
>
> [Houthooft et al., 2016] Rein Houthooft, Xi Chen, Xi Chen, Yan Duan, John Schulman, Filip De Turck, Pieter Abbeel, “VIME: Variational Information Maximizing Exploration,” NIPS 2016.
>
> [Kim et al., 2019]  Youngjin Kim, Wontae Nam, Hyunwoo Kim, Ji-Hoon Kim, and Gunhee Kim, “Curiosity-Bottleneck: Exploration by Distilling Task-Specific Novelty,” ICML 2019.
>
> [Badia et al., 2020] Adrià Puigdomènech Badia, Pablo Sprechmann, Alex Vitvitskyi, Daniel Guo, Bilal Piot, Steven Kapturowski, Olivier Tieleman, Martín Arjovsky, Alexander Pritzel, Andew Bolt, Charles Blundell, “Never Give Up: Learning Directed Exploration Strategies,” ICLR 2020.
>
> * (ii) Single-objective RL with constraints:
> In constrained RL problems, an RL agent is given additional cost signals (in addition to the reward signal), and the goal is to maximize total discounted reward subject to that the total discounted cost is kept below some budget [Achiam et al., 2017]. One approach is to consider the Lagrangian relaxation, which introduces the constraints into the RL objective as augmented cost signals with the help of Lagrange multipliers. In this way, we can reformulate the constrained RL problem as a multi-objective RL problem and view the Lagrange multipliers as the preference weights of the cost signals (this perspective has been mentioned in [Abdolmaleki et al., 2021]). From this perspective, we can again apply Q-Pensieve to learn a collection of policies (one for each preference vector) during training and thereafter select a policy that achieves high total reward and satisfies the cost constraints simultaneously.
>
> [Achiam et al., 2017] Joshua Achiam, David Held, Aviv Tamar, Pieter Abbeel, “Constrained Policy Optimization,” ICML 2017.
> [Abdolmaleki et al., 2021] Abbas Abdolmaleki, Sandy H. Huang, Giulia Vezzani, Bobak Shahriari, Jost Tobias Springenberg, Shruti Mishra, Dhruva TB, Arunkumar Byravan, Konstantinos Bousmalis, Andras Gyorgy, Csaba Szepesvari, Raia Hadsell, Nicolas Heess, Martin Riedmiller, “On Multi-objective Policy Optimization as a Tool for Reinforcement Learning,” arXiv Technical Report (arXiv:2106.08199), 2021.
>
> * (iii) Q-Ensemble RL algorithms:
> In addition to policy improvement, another use case of Q replay buffer is to facilitate the use of ensemble methods in RL. In ensemble RL algorithms, an RL agent would maintain a collection of Q networks (e.g., for enforcing exploration [Lee et al., 2021]), and Q replay buffer can be naturally integrated with this design.
>
> [Lee et al., 2021] Kimin Lee, Michael Laskin, Aravind Srinivas, Pieter Abbeel, “SUNRISE: A Simple Unified Framework for Ensemble Learning in Deep Reinforcement Learning,” ICML 2021.

---

### Official Review · Reviewer_ecaS · 2022-10-24

**Confidence:** 4
**Clarity, Quality, Novelty And Reproducibility:** 1. The update rule of $W_k(\lambda)$ …
**Correctness:** 3
**Technical Novelty And Significance:** 3
**Empirical Novelty And Significance:** 3
**Recommendation:** 6

**Strength And Weaknesses:**

Strength
1. The authors propose a novel multi-objective actor-critic algorithm inspired by Yang et al. (2019).
2. To improve sample efficiency, a Q-replay buffer is proposed, and it is efficiently integrated with the proposed algorithm.
3. The experimental results show that the proposed method performs better than the baselines with respect to several metrics.

Weakness
1. The sampling distribution is not discussed explicitly. If the preference is uniformly sample, applying the proposed method to problems with many objectives is difficult.
2. The scalability of the proposed method is not discussed.


**Summary Of The Paper:**

This paper proposes a method for improving sample efficiency in multi-objective reinforcement learning (MORL). Like universal value function approximators, a state-action value function represents a large number of value functions for any preferences. The most important contribution is the policy improvement step utilizing a set of previously learned Q-functions. The proposed method is evaluated on the MuJoCo benchmark tasks, and the experimental results show that the proposed method performs better in several metrics. Additionally, the proposed method outperforms a single-objective SAC because the proposed method can escape from sub-optimal policies.

**Summary Of The Review:**

Overall, this paper is well-written, and the experimental results are promising. However, the proposed method becomes more appealing if it is evaluated on problems with many objectives.

---

> ### Author Response · Authors · 2022-11-16
> **Response to Reviewer ecaS**
>
> We greatly appreciate the reviewer for the overall positive feedback and the detailed suggestions for improving the paper. We provide our response as follows:
>
> **(B1) Explain the update rule of $W_k(\lambda)$ and the sampling distribution in detail.**
>
> We describe the update rule of $W_k(\lambda)$ as follows. To ensure that the agent can learn well-performing policies for various preferences, in each iteration:
> Step 1: We first draw one preference vector (denoted by $\lambda$) randomly from some distribution $D$ for the actor update with respect to this $\lambda$. As also described in our response (C5), for tasks with $d$ objectives, we simply generate $d$ uniform random variables between zero and one and apply normalization such that the sum of them is one.
> Step 2: To form the preference set $W_k(\lambda)$, we simply draw another $M$ preference vectors in exactly the same way as in Step 1. In our experiments, we set $M=4$ without tuning.
> If one would like to optimize this part of the implementation, one possible way is to leverage the Dirichlet distribution (as done in CN-DER), which would provide a more "uniform" sampling distribution over the unit simplex.
>
> **(B2) The proposed method adopts the entropy-regularized framework, but the existing works (Abels et al., 2019; Yang et al., 2019) are based on the standard framework. Although I understand that policy improvement (8) is tightly linked to the entropy regularization of policy, I do not fully understand whether the proposed method requires the entropy regularization term.**
>
> Thank you for pointing out this. The technique of Q-Pensieve and Q replay buffer is quite generic (and could also benefit single-objective RL, as described in our response (C1)). We adopt the entropy-regularized framework to substantiate the proposed technique and thereby establish the nice convergence property of the resulting multi-objective soft policy iteration. Moreover, the standard unregularized RL framework can be readily recovered in the limit $\alpha \rightarrow 0$, as also mentioned in the SAC paper [Haarnoja et a., 2018].
>
> **(B3) Equations (1) and (5): V(s) should be V(s’).**
>
> Thank you for catching this. We have fixed the typo in the paper.

---

### Official Review · Reviewer_jdUh · 2022-11-01

**Confidence:** 3
**Correctness:** 3
**Technical Novelty And Significance:** 3
**Empirical Novelty And Significance:** 2
**Recommendation:** 6

**Clarity, Quality, Novelty And Reproducibility:**

The paper is overall clear. I have some clarification questions:
1. Do the actor and the critic network depend on the preference vector? If so, how is the preference vector handled as an input feature?

2. Can the authors provide more details on maintaining the Q replay buffer? How often did you push a new Q function into the buffer?

3. The practical implementation uses a relatively small Q replay buffer of size 4. Is the algorithm sensitive to this parameter?


**Strength And Weaknesses:**

Strength:
The Q-Pensieve idea is well-motivated as one can interpret this as an extension of the envelope Q-learning algorithm by Yang et. al. It is good that the Q-Pensieve retains the same convergence policy iteration results.

The empirical results are good as well, with Q-Pensieve outperforming other baselines in three different performance metrics. My understanding is that this is a fairly straightforward method to plug in so I appreciate its effectiveness.

Weakness:
The major weakness is the absence of the envelope Q-learning algorithm by Yang et. al. As the Q-Pensieve algorithm is closely related, the authors should provide a comparison.


**Summary Of The Paper:**

The paper presents Q-Pensieve, a method using versions of past Q functions to update a new Q function. This method enables information sharing across policies. The authors instantiate the idea in a soft actor-critic algorithm. Experiment results on DST, LunarLander, and several MuJoCo environments are provided where Q-Pensieve outperforms other baseline methods.

**Summary Of The Review:**

The proposed Q-Pensieve algorithm appears to be simple and effective. However, an important baseline is missing.

==========================
I want to thank the authors for writing a detailed response. My main concern, the comparison with envelope Q-learning, is adequately addressed. I would encourage the authors to expand the comparisons to include all the environments in a later revision.

On the Q replay buffer size study, it seems the performance is monotonically increasing with the buffer size. It would be good to add a discussion on the trade-off of the performance with memory overhead.

Overall, I decided to raise my score from 5 to 6.

---

> ### Author Response · Authors · 2022-11-16
> **Response to Reviewer jdUh**
>
> We greatly appreciate the reviewer’s constructive feedback for improving our paper. We provide our point-by-point response as follows.
>
> **(A1) Comparison with the Envelope Q-Learning algorithm**
>
> The Envelope Q-Learning algorithm and its neural version Envelope DQN presume that the action space is discrete. Accordingly, in [Yang et al., 2019], Enevelop DQN is only evaluated in RL tasks with discrete action space, such as discrete Deep Sea Treasure and SuperMario.
>
> To adapt Envelope DQN to the continuous control tasks considered in our paper (including MuJoCo and continuous Deep Sea Treasure), we take the open-source implementation in [Yang et al., 2019] and apply action discretization, which has been shown to be also quite effective in MuJoCo control tasks [Tavakoli et al., 2018; Tang and Agrawal, 2020].
>
> We compare Envelope DQN with Q-Pensieve in both Hopper and DST. For Envelope DQN, we set the number of bins for each action dimension to be 11 and 5 for DST (actions are 2-dimensional) and Hopper (actions are 3-dimensional), respectively, based on the suggestions provided by [Tavakoli et al., 2018]. The experimental results are provided in the table below:
>
> | Environments |Metrics | Envelope DQN  (1.5M steps) | Q-Pensieve  (1.5M steps) |
> | --------- | --------- | --------- | ---------
> | DST     | HV($\times10^{2}$)    | $6.83\pm0.13$     | **$10.21\pm1.40$** |
> |      |UT($\times10^{0}$)     | $4.43\pm0.05$     | **$7.31\pm0.91$** |
> | Hopper     | HV($\times10^{5}$)     | $0.38\pm0.19$     | **$13.31\pm2.03$** |
> |      | UT($\times10^{2}$)     | $-0.42\pm0.23$     | **$4.08\pm1.10$** |
>
> We can observe that Q-Pensieve outperforms Envelope DQN by a large margin in the above two popular multi-objective tasks.
>
> [Tavakoli et al., 2018] Arash Tavakoli, Fabio Pardo, Petar Kormushev, "Action Branching Architectures for Deep Reinforcement Learning," AAAI 2018.
>
> [Tang and Agrawal, 2020] Yunhao Tang, Shipra Agrawal, "Discretizing Continuous Action Space for On-Policy Optimization," AAAI 2020.
>
>
> **(A2) Do the actor and the critic networks depend on the preference vector? If so, how is the preference vector handled as an input feature?**
> Yes, the actor and the critic networks are designed to take the preference vector as part of the input. For the actor network, we concatenate the observation and the preference vector and make it the input of the actor network. Similarly, for the critic network, we concatenate the observation, action, and the preference vector to form the input of the critic network. Similar neural network architectures have also been used in several MORL methods, such as CN-DER and Envelope DQN.

---

### Author Response · Authors · 2022-11-16
**Response to All Reviewers**

**(All-3) Provide more details on the implementation of Q replay buffer and explain how often a new Q function is pushed into the buffer.**

To implement the Q-Pensieve policy improvement, we store the past critic networks in a Q replay buffer in a FIFO manner, and we use these Q snapshots to assist the policy update.
Regarding the size of the Q replay buffer, we set it to be 4 in all the experiments in the original manuscript. Moreover, based on our response to (All-2), we find that empirically a relatively small Q buffer size already offers a significant performance improvement.
Furthermore, in our experiments, we also found that the memory footprint of the Q replay buffer is rather small. For example, for the Ant in MuJoco, each Q snapshot (with two hidden layers, hidden size = 256) takes about only 15MB of GPU memory.

Regarding the update interval of the Q buffer, we push a new Q snapshot into the Q buffer every 1000 steps for all the experiments in the paper. Notably, we would like to highlight that **this update interval has negligible effect on the GPU memory usage and computation time as the pop/push operations of a FIFO only takes minimal computation time and these operations do not take extra memory**.

On the other hand, to ensure that the Q snapshots in the Q replay buffer are rather diverse, we would suggest that the update interval shall not be too small (otherwise the Q snapshots in the buffer would be fairly similar). Moreover, as in general this update interval can be viewed as a hyperparameter to be tuned (similar to the update interval of the target networks in many RL algorithms), we further do an empirical study on the performance of Q-Pensieve under different update intervals. We run Q-Pensieve with different update intervals in Ant for 700k steps. The experimental results are provided in the table below:


| Update Interval (steps) | 500 | 1000 | 1500 | 2000 |
| -------- | -------- | -------- |-------- | -------- |
| HV ($\times 10^{6}$)     | 6.59    | 7.47    |6.89     | 6.87     |
| UT ($\times 10^{2}$)    | 2.64     | 3.00     |2.94     | 4.90|

We can observe that the hypervolume is not sensitive to the update interval, and the performance of Q-Pensieve can potentially be improved through hyperparameter tuning.

---

### Author Response · Authors · 2022-11-16
**Response to All Reviewers**

We thank all the reviewers for their valuable feedback and insightful suggestions!

For ease of discussion, we first provide our response to those questions asked by multiple reviewers in this thread. Then, we post our responses to the rest of the questions right below the review comments of each individual reviewer.

**(All-1) Discuss the scalability of the proposed method, especially the ability to deal with many objectives.**

Thank you for the suggestion. First, we would like to clarify that in our original manuscript, Q-Pensieve and the benchmark methods have already been evaluated in tasks with **more than 2 objectives**, including LunarLander (4 objectives), Hopper3d (3 objectives), and Ant3d (3 objectives), and the detailed description about the physical meaning of the objectives is provided in Appendix B.1 in the original manuscript.
To make this more clear, we have also edited the description of Table 1 in the paper.

Moreover, to better demonstrate the scalability of Q-Pensieve in terms of the number of objectives, we further evaluate the algorithms in the following two tasks with 5 objectives:
* Hopper5d: The 5 objectives are forward speed, control cost of each of the 3 joints, and healthy reward.
* LunarLander5d: The 5 objectives are the result reward, shaping reward, main engine cost, side engine cost, and time penalty.

Notably, in the original papers of the benchmark methods like CN-DER [Abels et al., 2019] and PGMORL [Xu et al., 2020], the algorithms are evaluated in tasks with only at most 3 objectives.

The experimental results are shown below:



| Environments |Performance Metrics | PGMORL  (1.5M steps) | PGMORL  (7.5M steps) | CN-DER  (1.5M steps) | Q-Pensieve  (1.5M steps) |
| --------- | --------- | --------- | --------- | --------- | --------- |
|      | HV($\times10^{13}$)     | $0.60\pm0.08$     | $0.44\pm0.93$ | $3.03\pm0.81$     | **$7.15\pm0.29$**     |
| Hopper5d     | UT($\times10^{2}$)     | $1.50\pm0.26$     | $1.56\pm0.18$ | $2.43\pm0.33$     | **$3.38\pm0.64$**     |
|      | ED     | $0.18\pm0.08$     | $0.15\pm0.04$ | $0.20\pm0.05$     | **$0.53\pm0.05$**     |
|      | HV($\times10^{11}$)     | $1.82\pm0.17$     | $1.91\pm0.44$ | $8.64\pm0.15$     | **$9.48\pm1.84$**    |
| LunarLander5d     | UT($\times10^{1}$)     | $-2.97\pm0.66$     | $-4.37\pm1.00$ | $0.63\pm0.39$     | **$1.20\pm0.22$**    |
|      | ED     | $0.06\pm0.02$     | $0.05\pm0.03$ | $0.49\pm0.01$     | **$0.51\pm0.02$**    |

We can observe that Q-Pensieve still significantly outperforms the benchmark methods in terms of the multi-objective performance metrics. On the other hand, we also find that PGMORL has very little learning progress even after 7.5M steps, and this appears reasonable as it would be more difficult to apply explicit search in problems with more objectives.

**(All-2) The choice of Q buffer size.**

Based on the Q-Pensieve Policy Improvement in Eq. (8), one could expect that a larger Q buffer size could help provide a more diverse collection of Q snapshots and thereby better boost the policy improvement update in each iteration. On the other hand, in practice, the required memory usage also scales with the Q buffer size.
Despite this, we found that empirically a relatively small Q buffer size already offers a significant performance improvement.
To see this, we evaluate Q-Pensieve under buffer sizes = 2, 4, 6 and compare it to that without using a Q replay buffer, in the Ant environment of MuJoCo.



|  | Without Q Buffer | Buffer Size = 2 | Buffer Size = 4 | Buffer Size = 6 |
| -------- | -------- | -------- |-------- | -------- |
| HV($\times10^{7}$)     |  $0.91\pm0.26$    | $0.97\pm0.23$     |$1.04\pm0.25$ | $1.25\pm0.15$ |
| UT($\times10^{2}$)    | $8.23\pm0.48$    | $12.48\pm0.28$     |$14.02\pm0.26$ | $15.67\pm0.33$  |

As shown in the table above, we can indeed observe that (i) the performance of Q-Pensieve increases with the Q buffer size, and (ii) the performance gap between Q-Pensieve and that without a Q replay buffer is already rather significant under a Q buffer size = 2.

---

### Author Response · Authors · 2022-12-02
**Authors' Response**

We thank all the reviewers for the valuable and helpful suggestions, and we hope that our responses have answered all the questions. If possible, would the reviewers kindly let us know if our replies have well addressed the questions? We will be delighted to respond accordingly if there are any further comments. Thank you.

---

### Decision · Program_Chairs · 2023-01-20

**Decision:**

Accept: poster

**Justification For Why Not Higher Score:**

Although the paper presents nice contributions, the idea is not really novel, and it is not clear how this approach scales to more complex domains.

**Justification For Why Not Lower Score:**

The paper deals with a relevant topic and proposes an interesting and effective idea.
There are some weak points, but most of them were properly addressed by the authors in their rebuttals.

**Metareview: Summary, Strengths And Weaknesses:**

The paper presents a value-based method for Multi-Objective Reinforcement Learning problems that uses past Q-functions to update the new Q-function.
The idea is implemented in an actor-critic algorithm, which is evaluated in several environments showing better performance than some baselines.
At the end of the reviewing period, this paper was borderline, with some reviewers voting for acceptance and others for rejection.
The authors' feedback effectively solved most of the reviewers' concerns. After a round of discussion, the reviewers agree that the merits of this paper outweigh the defects.
The authors must update their paper following the reviewer's suggestions carefully while preparing the final version of their paper.

**Note From Pc:**

if the above contains the word "oral" or "spotlight" please see: "oral" presentation means -> notable-top-5% and "spotlight" means -> notable-top-25%. As stated in our emails, we are disassociating presentation type from AC recommendations

**Summary Of Ac-Reviewer Meeting:**

Only two reviewers attended the virtual meeting (VKcR and jdUh).
Reviewer jdUh had some concerns about the choice of the baselines and the lack of motivation for some specific choices.
These concerns were well addressed by the authors, and so the reviewer has changed his mind about this paper.
Reviewer VKcR was really positive about the paper; he appreciated the idea, the empirical results are very strong, and there is also an effort to provide some theoretical guarantee.
Together with these two reviewers, we gave a look at the concerns of the other two reviewers, and we found they were well addressed by the authors,
For these reasons, I have decided to accept this paper.